# Beyond Trajectory-Level Attribution: Graph-Based Credit Assignment for Agentic Reinforcement Learning

Xin Cheng[1]  Shuo He[1]  Lang Feng[1]  HaiYang Xu[2]  Ming Yan[2]  Lei Feng[3]  Bo An[1]

## Abstract

Group-based reinforcement learning (RL) methods have achieved remarkable success in improving the performance of large language models (LLMs) and have been rapidly extended to agentic tasks. However, their credit assignment relies heavily on coarse-grained trajectory-level attribution according to final outcomes, making it difficult to capture the contribution of individual steps, such as valuable steps obscured within failed trajectories. *To uncover latent information and enable more faithful step-level credit assignment*, we propose Graph-based Group Policy Optimization (GraphGPO), which first aggregates all rollout trajectories into a unified state-transition graph and then estimates the distance from each state to the task goal using the global information encoded in the graph. Finally, GraphGPO assigns credit to each edge by estimating a graph-based advantage, based on how much the transition reduces the distance to the task goal. In this way, GraphGPO significantly improves training efficiency and achieves state-of-the-art performance across a range of challenging benchmarks. Code is available at https://github.com/langfengQ/ver l-agent/tree/master/recipe/GraphGPO.

## 1. Introduction

In recent years, Large Language Models (LLMs) (Achiam et al., 2024; Gemini et al., 2023; Yang et al., 2025; Liu et al., 2025a) have undergone rapid iteration and evolution, enabling them to move beyond static language understanding (Devlin et al., 2019; Radford et al., 2019) toward more complex reasoning and decision-making tasks (Wei et al., 2022; Ahn et al., 2022). As a result of these advances,

LLMs are increasingly deployed as agents that can perceive, reason, and act in complex, open-ended environments, enabling them to tackle tasks that require long-horizon planning and sequential decision making, spanning embodied tasks (Wang et al., 2023; Driess et al., 2023), web or mobile interactive environments (Furuta et al., 2024; Wang et al., 2024a; Zheng et al., 2024a; Gou et al., 2025; Feng et al., 2025a), as well as interactive games (Hafner et al., 2023; Xu et al., 2024; Wang et al., 2025d; Liu et al., 2025b) and tool-augmented reasoning scenarios (Schick et al., 2023; Paranjape et al., 2023; Qian et al., 2025; Xue et al., 2025).

Reinforcement Learning (RL) (Sutton et al., 1998) has demonstrated its ability to drive agents toward human-level performance through landmark achievements (Silver et al., 2018). As learning paradigms evolved toward large-scale models, RL has re-emerged as a crucial post-training stage for LLMs to enhance performance, resulting in frontier models such as OpenAI o1 (OpenAI, 2024) and DeepSeek R1 (Guo et al., 2025). Notably, group-based RL methods (Yu et al., 2025b; Zheng et al., 2025a; Cui et al., 2025), such as GRPO (Shao et al., 2024), have gained prominence by discarding the resource-intensive critic model used in traditional Actor-Critic frameworks (Konda & Tsitsiklis, 1999; Schulman et al., 2015a; 2017; Haarnoja et al., 2018). These methods rely on verifiable rewards and intra-group statistics, which reduces memory consumption and enables efficient scaling to LLMs. More recently, several studies (Wang et al., 2025e; Jin et al., 2025; Luo et al., 2025a; Yu et al., 2025a; Wang et al., 2025c; Feng et al., 2025b; He et al., 2026) have begun to extend group-based RL to multi-turn agentic tasks.

Although existing group-based RL methods have shown promising performance in multi-turn agentic tasks, they rely on an implicit but restrictive assumption: *the quality of each step can be inferred solely from the final success or failure of the trajectory it belongs to.* Under this assumption, all steps within a successful trajectory receive positive credit, while all steps within a failed trajectory are penalized, regardless of their actual contribution to task progress. However, this trajectory-level attribution is fundamentally misaligned with the multi-turn agentic tasks. A successful trajectory may contain redundant or erroneous steps that do not advance the agent toward the goal, while a failed trajectory may in-

[1]Nanyang Technological University, Singapore [2]Tongyi Lab, Alibaba Group [3]Southeast University, China. Correspondence to: Lei Feng <fenglei@seu.edu.cn>.

*Proceedings of the 43rd International Conference on Machine Learning*, Seoul, South Korea. PMLR 306, 2026. Copyright 2026 by the author(s).

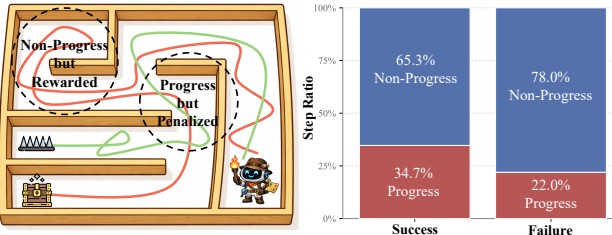

*Figure 1.* **Left:** When one successful trajectory and one failed trajectory are sampled, non-progress steps within the successful trajectory receive positive credit, while progress steps within the failed trajectory are penalized. **Right:** Step statistics showing the proportion of progress and non-progress steps in early-stage training of ALFWorld (rollout $M = 8$ and maximum step $T = 50$), reported separately for successful and failed trajectories.

clude decisive and valuable steps whose effects are negated by later mistakes. Figure 1 presents step-level statistics, showing that approximately 22.0% of the steps in failed trajectories contribute to task progress, whereas about 65.3% of the steps in successful trajectories do not meaningfully advance the task. As a result, credit assignment that relies solely on trajectory-level attribution fails to capture such latent information, leading to coarse and noisy step-level signals and, consequently, inefficient learning.

To address these problems, we propose Graph-based Group Policy Optimization (GraphGPO), a novel RL method that departs from trajectory-level attribution dependency and instead aggregates all rollout trajectories into a unified state-transition graph, enabling global, structure-aware finer-grained credit assignment. Concretely, GraphGPO represents environment states (e.g., the current prompt or historical interactions) as nodes, and models each step across all trajectories as a directed edge, thereby replacing isolated trajectory representations with a unified dynamic state-transition graph. Leveraging the global information from the graph, GraphGPO then captures the relationship between each state and the task goal to assign step-level rewards, without relying on trajectories. Finally, outgoing edges originating from the same state are grouped together to estimate graph-based advantages, which are then used to optimize the agent. Compared with previous methods, GraphGPO leverages global experiential information aggregated from all trajectories to capture latent relationships among steps, enabling more faithful step-level credit assignment and significantly improving training efficiency. Moreover, GraphGPO remains critic-free while introducing only a negligible amount of graph computation, achieving stable convergence without incurring additional overhead.

## 2. Related Work

**Reinforcement learning for LLMs.** Early applications of RL to LLMs primarily focused on RL from human feed-

back (RLHF) (Ziegler et al., 2019; Stiennon et al., 2020; Ouyang et al., 2022; Bai et al., 2022; Rafailov et al., 2023; Zhang et al., 2025), where human preferences are used to align LLMs. More recently, RL with verifiable rewards (RLVR) (Kool et al., 2019; Ahmadian et al., 2024; Shao et al., 2024; Liu et al., 2025c; Lin et al., 2025; Su et al., 2025) has gained increasing attention, replacing human feedback with automatically computable rewards. This paradigm has been shown effective for improving reasoning capabilities (Team et al., 2025; Guo et al., 2025; Wen et al., 2025) in domains such as mathematics (Lightman et al., 2023), code generation (Le et al., 2022; Sun et al., 2024; Jiang et al., 2025), tool use (Schick et al., 2023; Qin et al., 2023; Wang et al., 2025a; Qian et al., 2025) and search (Sun et al., 2025; Song et al., 2025; Zheng et al., 2025b).

**Reinforcement Learning for LLM-based Agents.** Beyond static text generation, RL has increasingly been used to enhance the capabilities of LLM-based agents in dynamic, open-ended environments. Early studies (Mnih et al., 2015; Tan et al., 2024; Wen et al., 2024; Zhai et al., 2024; Bai et al., 2024; Wang et al., 2024c) typically employ value-based critics (Schulman et al., 2017; Peng et al., 2019) to guide agent learning in domains such as Android device control (Rawles et al., 2023), embodied environments (Shridhar et al., 2021), and card games (Brockman et al., 2016). More recent work has begun to extend RL to real-world agentic settings, including software engineering (Yang et al., 2024; Zheng et al., 2024b; Da et al., 2025), GUI control (Lu et al., 2025; Wang et al., 2025b; Lai et al., 2025; Shi et al., 2025), and Model Context Protocol (Luo et al., 2025b; Le et al., 2025; Team & Team, 2025).

## 3. Preliminaries

In this section, we introduce preliminary knowledge of multi-turn agentic tasks, group-based reinforcement learning, and relevant advantage estimation.

### 3.1. Multi-turn Agentic Tasks

Let $x \sim p(X)$ be the task example, and let $\pi_\theta$ denote a LLM-based policy parameterized by $\theta$. In the general agentic setting (Xi et al., 2025; Wang et al., 2024b), the policy needs to interact with the environment multiple steps to accomplish the goal associated with task $x$. At each step $t = 1, 2, \ldots, T$, the policy $\pi_\theta$ observes the current environment state $s_t$ and produces an action $a_t \in \mathcal{V}^n$ sampled from conditioned distribution $\pi_\theta(a_t \mid s_t, x)$, where $\mathcal{V}$ denotes the token vocabulary and $n$ is the allowed maximum generation length. The environment then transitions to the next state $s_{t+1}$. Ultimately, all interactions form a trajectory $\tau = \{(s_1, a_1), (s_2, a_2), \ldots, (s_T, a_T)\}$. In many real-world tasks, the reward is provided only at the end of the trajectory depending on whether the goal is successfully accomplished,

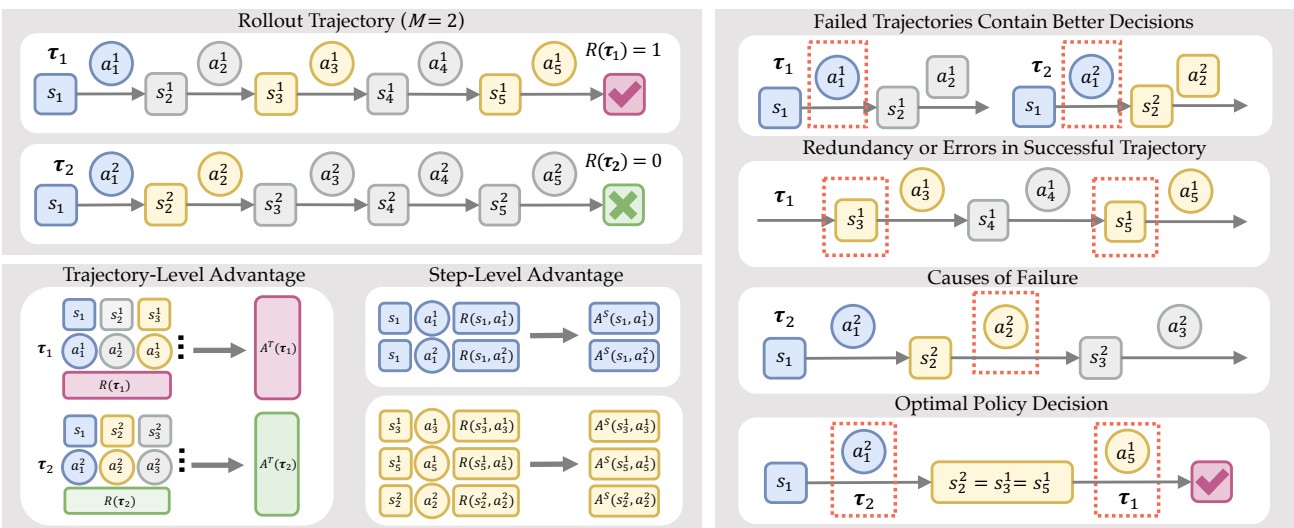

*Figure 2.* Overview of group-based advantage estimation and existing issues, where squares represent states and circles represent actions. **Top-left:** Rollout trajectories ($\tau_1$ is a successful trajectory and $\tau_2$ is a failed trajectory), where blue squares denote identical states among themselves, yellow squares denote another set of identical states, and gray represents independent states with no shared states. **Bottom-left:** Trajectory-level and step-level advantage estimation. **Right:** Issues with credit assignment that relies solely on trajectory success in both successful and failed trajectories.

so most intermediate rewards are zero. In this paper, we focus on such sparse, delayed reward settings.

### 3.2. Group-based Reinforcement Learning

The advantage function ([Schulman et al., 2015b](#); [Wang et al., 2016](#)) in RL plays a central role by quantifying how favorable an action is and guiding policy updates. Group-based RL methods, such as Group Relative Policy Optimization (GRPO) ([Shao et al., 2024](#)), estimate advantages in a critic-free manner by relying on verifiable rewards and intra-group statistics, thereby simplifying the training architecture. More formally, for a task instance $\boldsymbol{x}$, a group of outputs $G^o = \{o_1, o_2, \ldots, o_M\}$ is sampled from the policy model $\pi_\theta$. Each output $o_m$ receives a scalar reward $R(o_m)$ based on whether the goal of task is successfully completed by $o_m$. GRPO then estimates advantages solely from statistics computed within the group:

$$A(o_m) = (R(o_m) - \mu(G^o))/\sigma(G^o), \quad (1)$$

where $\mu(\boldsymbol{o})$ and $\sigma(\boldsymbol{o})$ denote the mean and standard deviation of rewards $R(o)$ within the group $G^o$. GRPO was originally designed for single-turn settings, but it can be readily extended to multi-turn agentic tasks:

$$A^E(\boldsymbol{\tau}_m) = (R(\boldsymbol{\tau}_m) - \mu(G^E))/\sigma(G^E), \quad (2)$$

where $G^E = \{\boldsymbol{\tau}_1, \boldsymbol{\tau}_2, \ldots, \boldsymbol{\tau}_M\}$ denotes a group of trajectories, and $R(\boldsymbol{\tau}_m)$ is a scalar reward depending on whether the trajectory can successfully accomplish the goal.

For each step $(s_t^m, \boldsymbol{a}_t^m)$ within the same trajectory $\boldsymbol{\tau}_m$, GRPO assigns the same reward $R(\boldsymbol{\tau}_m)$ and episode-level

advantage $A^E(\boldsymbol{\tau}_m)$, which results in overly coarse credit assignment. To address fine-grained credit assignment, Group-in-Group Policy Optimization (GiGPO) ([Feng et al., 2025b](#)) introduces a step-level group advantage estimator which groups together all steps with the same state, regardless of whether they come from the same trajectory. The step-level group advantage is then computed as follows:

$$A^S(s_t^m, \boldsymbol{a}_t^m) = (R^S(s_t^m, \boldsymbol{a}_t^m) - \mu(G^S(s_t)))/\sigma(G^S(s_t)),$$

where $G^S(s_t) = \{(s_i^j, \boldsymbol{a}_i^j) \mid s_i^j = s_t, \ 1 \le i \le T, \ 1 \le j \le M\}$ denotes the group of steps that share the same initial state $s_t$, and $R^S(s_t^m, \boldsymbol{a}_t^m) = \lambda^{T-i} R(\boldsymbol{\tau}_m)$ represents the standard RL discounted reward. Although the step-level credit assignment in GiGPO improves agent performance, it still relies on trajectory-level attribution based on the final outcome, i.e., $R(\boldsymbol{\tau})$.

## 4. Proposed Method

In this section, we first discuss the limitations of existing group-based RL methods. We then introduce a novel and effective method that enables faithful step-level credit assignment without incurring additional overhead.

### 4.1. Limitations of Trajectory-Level Attribution

Although existing group-based RL methods have employed multi-level advantage estimation and provide effective training signals in agentic tasks, their credit attribution still fundamentally relies on trajectory-level outcomes, limiting the fidelity of step-level credit assignment.

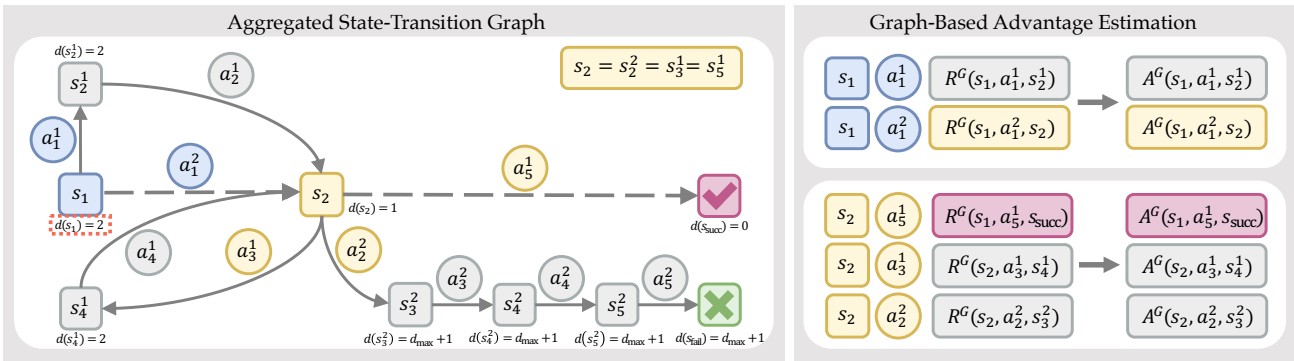

*Figure 3.* Overview of GraphGPO. For simplicity, we assume all transition costs are unitary, i.e., $c(s, \boldsymbol{a}) = 1$. **Left:** The aggregated state-transition graph constructed based on states from rollout trajectories, where identical states are merged (e.g., $s_1 = s_1^2 = s_2^1 = s_1^4$). **Right:** Graph-based advantage estimation, where the credit assignment relies on the the distance $d(\cdot)$ of the next state. Taking the initial state $s_1$ as an example, the shortest path to the goal state $s^s$ is $\{(s_1, \boldsymbol{a}_1^2, s_2), (s_2, \boldsymbol{a}_5^1, s_{\text{succ}})\}$, yielding $d(s_1) = 2$. Moreover, the maximum shortest-step distance among all states that can reach the goal is $d_{\max} = 2$.

Specifically, current methods assign credit by attributing step quality solely to the final success or failure of a trajectory. As a result, redundant or erroneous steps within successful trajectories are rewarded, while correct steps in failed trajectories are consistently penalized. This attribution misalignment not only impedes effective policy updating but also obscures the underlying reasons for trajectory failures. Figure 2 illustrates these issues. For trajectory $\boldsymbol{\tau}_1$, although the task is ultimately completed successfully, repeatedly visiting the yellow states ($s_3^1$ and $s_5^1$) indicates erroneous and redundant behavior. In contrast, although trajectory $\boldsymbol{\tau}_2$ eventually fails, it takes a more favorable action $\boldsymbol{a}_2^1$ than $\boldsymbol{\tau}_1$ at the blue state $s_1$, enabling the environment to reach the yellow state more quickly. Moreover, the failure of $\boldsymbol{\tau}_2$ may stem from the action $\boldsymbol{a}_2^2$ in the yellow state $s_2^2$, which leads to a successor state $s_3^2$ that cannot reach the goal state $s_{\text{succ}}$. Therefore, the steps preceding this state should not be directly penalized.

To address these issues, we propose Graph-based Group Policy Optimization (GraphGPO), a novel graph-based credit assignment method that leverages the global state-transition structure to redefine step-level credit attribution, thereby providing more faithful guidance for policy optimization.

### 4.2. The Aggregated State-Transition Graph

Rather than treating each trajectory independently, we aggregate all rollout trajectories into a unified state-transition graph, which provides a structured representation of the environment dynamics induced by interactions of the policy. This representation enables credit assignment based on the global connectivity between states. Formally, given a set of rollout trajectories $\{\boldsymbol{\tau}_1, \boldsymbol{\tau}_2, \ldots, \boldsymbol{\tau}_M\}$ of size $M$, we construct a directed graph $\mathcal{G} = (\mathcal{S}, \mathcal{E})$. The node set $\mathcal{S}$ consists of all states visited in the trajectories, i.e., $\mathcal{S} = \bigcup_{m=1}^{M} \bigcup_{t=1}^{T} \{s_t^m\}$. For states that are identical, we merge

them into a single node in the graph. A directed edge $(s, \boldsymbol{a}, s', c(s, \boldsymbol{a})) \in \mathcal{E}$ exists if and only if there exists a time step $t$ and a trajectory $\boldsymbol{\tau}_m$ containing a transition $\{\ldots (s_t^m, \boldsymbol{a}_t^m), (s_{t+1}^m, \boldsymbol{a}_{t+1}^m) \ldots\}$ such that $s_t^m = s$, $\boldsymbol{a}_t^m = \boldsymbol{a}$, and $s_{t+1}^m = s'$, where $c(s, \boldsymbol{a}) > 0$ denotes the cost incurred when the policy takes action $\boldsymbol{a}$ at state $s$. In tool-use scenarios, the cost can be defined as the cost associated with tool usage, such as monetary or time costs. We denote the set of terminal states as $\mathcal{S}_{\text{term}}$, which includes both goal states $s_{\text{succ}}$ corresponding to task successful completion and states $s_{\text{fail}}$ which exceeds the maximum allowed length. In Figure 3, we show the aggregated state-transition graph constructed based on the successful trajectory $\boldsymbol{\tau}_1$ and failed trajectory $\boldsymbol{\tau}_2$ from Figure 2.

### 4.3. The Graph-Based Advantage Estimation

The graph-based structured representation provides global dynamic information between states. For example, it allows us to easily identify which states can be reached from any given state, and even how to reach them, even if it requires traversing multiple trajectories. Moreover, it also allows us to identify states from which cannot reach the goal state $s_{\text{succ}}$. These states should be avoided. Most importantly, it enables us to determine which states are closer to goal state $s_{\text{succ}}$. Building on this idea, for any state $s$, we use the following distance to measure the minimum cost required to reach the goal state $s_{\text{succ}}$:

$$d(s) = \begin{cases} 0, & \text{if } s = s_{\text{succ}}, \\ \min_{(s,a,s',c)\in\mathcal{E}} \big(c(s, \boldsymbol{a}) + d(s')\big), & \text{if } s \rightsquigarrow s_{\text{succ}}, \\ +\infty, & \text{otherwise}, \end{cases} \quad (3)$$

where $s \rightsquigarrow s_{\text{succ}}$ denotes that there exists a path from $s$ to the goal state $s_{\text{succ}}$ in the graph. If $d(s)$ is close to 0, it indicates that state $s$ is close to successfully completing the task. If $d(s)$ is large, it implies that significant cost is required to

complete the task. When $d(s) = +\infty$, it indicates that the state is empirically unable to complete the task, which indicates failure. Naturally, a good action is expected to reduce the distance to the goal while incurring minimal cost. Therefore, for an edge $(s, \boldsymbol{a}, s', c(s, \boldsymbol{a})) \in \mathcal{E}$ in the graph $\mathcal{G}$, we define the following graph-based step-level reward:

$$R^G(s, \boldsymbol{a}, s') = r_{\text{succ}} \, \omega^{d(s')+c(s,\boldsymbol{a})}, \qquad (4)$$

where $r_{\text{succ}} > 0$ is a scalar denoting the reward for successfully completing the task, which is typically set to 1 or 10 in general agentic settings. $\omega \in (0, 1)$ is a distance discount factor that discounts the reward $r_{\text{succ}}$ according to the distance between the resulting state $s'$ and the goal. Additionally, for states with $d(s') = +\infty$, to avoid overly strong penalties that may restrict policy exploration, we replace $+\infty$ with the largest finite distance in the graph plus one, i.e., $d_{\max} + 1$, where $d_{\max} = \max_{s \in \mathcal{S}} d(s)$. The graph-based step-level reward discards feedback based on individual trajectories, and instead leverages the global connectivity of the aggregated graph constructed from all trajectories to provide more faithful step-level credit assignment. We then define the step-level group for each state $s$ in graph as:

$$G^G(s) = \{(s_i, \boldsymbol{a}, s_j) \mid (s_i, \boldsymbol{a}, s_j) \in \mathcal{E}, \ s_i = s\}. \qquad (5)$$

Intuitively, this group represents all candidate state transitions that originate from state $s$, i.e., all edges in the graph that start from $s$. Once this group is formed, we can estimate the following graph-based advantage for each edge $(s, \boldsymbol{a}, s') \in \mathcal{E}$ in the graph $\mathcal{G}$ :

$$A^G(s, \boldsymbol{a}, s') = \frac{R^G(s, \boldsymbol{a}, s') - \mu(G^G(s))}{\sigma(G^G(s))}, \qquad (6)$$

where $\mu(G^G(s))$ and $\sigma(G^G(s))$ denote the mean and standard deviation of graph-based step-level rewards within the group $G^G(s)$. For each state $s$ with a step-level group size $|G^G(s)| = 1$, i.e., the state appears only once in collected experience trajectories, the graph-based advantage $A^G(s, \boldsymbol{a}, s')$ is set to 0.

Through graph-based advantage estimation, we derive step-level credit from global state-transition structure, enabling more faithful credit attribution. Specifically, for the start state $s_1$ in Figure 3, it has two successor states $s_2^1$ and $s_2$. Since state $s_2$ is closer to the goal state $s_{\text{succ}}$ than state $s_2^1$ (i.e., $d(s_2) < d(s_2^1)$), the resulting advantage satisfies $A^G(s_1, \boldsymbol{a}_1^2, s_2) > A^G(s_1, \boldsymbol{a}_1^1, s_2^1)$. This indicates that action $\boldsymbol{a}_1^2$ is preferred over $\boldsymbol{a}_1^1$ at state $s_1$, even though $\boldsymbol{a}_1^2$ comes from the failed trajectory $\boldsymbol{\tau}_2$. In addition, we can identify action $\boldsymbol{a}_1^2$ as the cause of the failure, since it transitions to state $s_2^2$, which empirically has no path to the goal state $s_{\text{succ}}$ (i.e., $d(s_3^2) = d_{\max} + 1$). As a result, the corresponding advantage $A^G(s_2, \boldsymbol{a}_2^2, s_3^2)$ is the smallest within the group $G^G(s_2)$. Moreover, redundancy or erroneous behavior is reflected as cycles in the graph, such as the loop

$(s_2, \boldsymbol{a}_3^1, s_4^1), (s_4^1, \boldsymbol{a}_4^1, s_2)$ in Figure 3. Such transitions inevitably increase the distance $(d(s_4^1) > d(s_2))$ to the goal state $s_{\text{succ}}$ and thus receive lower advantages than actions that reduce the distance, e.g., $\boldsymbol{a}_5^1$. We then present the following proposition, which demonstrates that graph-based advantage estimation assigns higher advantages to actions that more effectively reduce the distance to the goal state.

**Proposition 4.1** (Monotonicity of graph-based advantage). *Consider the state $s$ that can reach the goal in the deterministic environment. There are two actions $a_{good}$ and $a_{bad}$ leading state $s$ to next states $s'_{good}$ and $s'_{bad}$, respectively. Assume that $a_{good}$ moves closer to the goal than $a_{bad}$, i.e., $d(s'_{good}) + c(s, a_{good}) < d(s'_{bad}) + c(s, a_{bad})$. The proposed graph-based advantage satisfies*

$$A^G(s, a_{good}, s'_{good}) \ > \ A^G(s, a_{bad}, s'_{bad}). \qquad (7)$$

The proof of Proposition 4.1 is provided in Appendix B.1. In particular, credit attribution based on final trajectory outcomes favors actions that merely appear in successful trajectories, whereas graph-based credit attribution prefers actions that move closer to the goal.

### 4.4. Policy Optimization

To prevent the advantage from degenerating to zero for states with only one outgoing edge (i.e., $|G^G(s)| = 1$), we still incorporate the episode-level advantage, resulting in the following combined advantage:

$$A(s, \boldsymbol{a}, s') = \beta^G A^G(s, \boldsymbol{a}, s') + \beta^E A^E(\boldsymbol{\tau}), \qquad (8)$$

where the edge $(s, \boldsymbol{a}, s')$ belongs to the trajectory $\boldsymbol{\tau}$, meaning that there exists a time step $t$ such that $\boldsymbol{\tau}$ contains the transition $\{\dots (\boldsymbol{s}_t, \boldsymbol{a}_t), (\boldsymbol{s}_{t+1}, \boldsymbol{a}_{t+1}) \dots\}$ with $\boldsymbol{s}_t = s$, $\boldsymbol{a}_t = \boldsymbol{a}$, and $\boldsymbol{s}_{t+1} = s'$. $\beta^G$ and $\beta^E$ are balancing factors for the advantages $A^G$ and $A^E$, respectively. Then the final policy optimization objective of GraphGPO is:

$$\mathbb{E}\left[ \frac{1}{NT} \sum_{m=1}^{M} \sum_{t=1}^{T} \min\Big( \rho_\theta(\boldsymbol{a}_t^m) A(s_t^m, \boldsymbol{a}_t^m, s_{t+1}^m), \qquad (9) \right.$$

$$\left. \text{clip}\big(\rho_\theta(\boldsymbol{a}_t^m), 1 \pm \epsilon\big) A(s_t^m, \boldsymbol{a}_t^m, s_{t+1}^m)\Big) \right] - \beta \mathbb{D}_{\text{KL}}\big(\pi_\theta \,\|\, \pi_{\theta_{\text{ref}}}\big),$$

where $\rho_\theta(\boldsymbol{a}_t^m) = \frac{\pi_\theta(\boldsymbol{a}_t^m | s_t^m, x)}{\pi_{\text{old}}(\boldsymbol{a}_t^m | s_t^m, x)}$ is the importance sampling ratio and $\beta$ controls the strength of the KL penalty. We present the pseudo code in Appendix A.

Compared to previous methods, GraphGPO provides more faithful step-level credit assignment by deriving credit from the global state-transition structure rather than trajectory outcomes. We then present the following proposition, which shows that the conditional variance of graph-based step-level feedback is lower than that of trajectory-based step-level feedback.

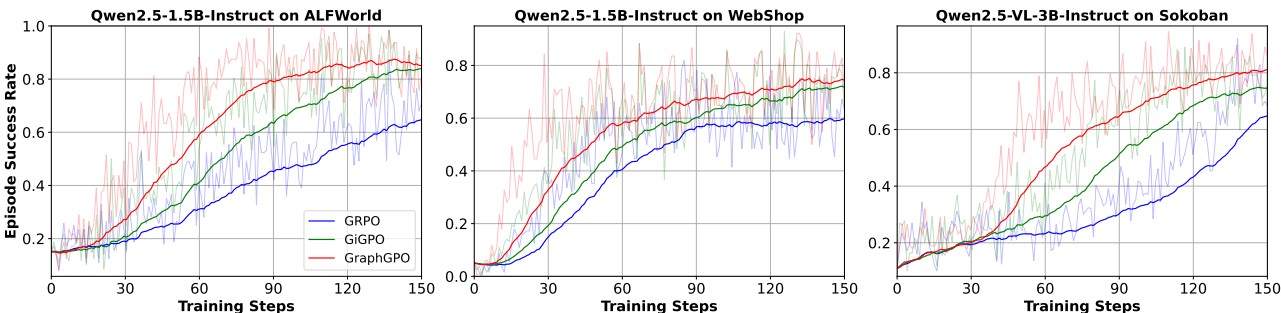

*Figure 4.* Training episode success rate versus steps for GraphGPO (red), GiGPO (green), and GRPO (blue) on the ALFWorld, WebShop, and Sokoban benchmarks. The lighter curves show the original curves, while the darker curves correspond to exponential moving average (EMA) smoothing with decay $\alpha = 0.95$, highlighting the overall training trends.

**Proposition 4.2** (Conditional variance reduction). *Given rollouts sampled from a fixed policy $\pi$ in the deterministic environment. For a state–action pair $(s, \boldsymbol{a}, s')$ that is visited with non-zero probability, Assume that both successful and failed trajectories pass through $(s, \boldsymbol{a}, s')$ with positive probability and $(s, \boldsymbol{a}, s')$ is independent of step $t$. For the trajectory-style step feedback $X^S = \eta(t)R(\boldsymbol{\tau})$ and graph-style step feedback $X^G = R^G(s, \boldsymbol{a}, s)$, the conditional variance hold almost surely:*

$$Var(X^G|s, a, s') \leq Var(X^S|s, a, s').$$

*where $\eta(t) = \lambda^{T-t}$ for GiGPO, $\eta(t) = 1$ for GRPO.*

The proof of Proposition 4.2 is provided in Appendix B.2.

## 5. Experiments

In this section, we present extensive experiments to demonstrate the effectiveness of our proposed method.

### 5.1. Experimental Setup

**Benchmarks.** We evaluate GraphGPO on a set of challenging multi-turn agentic benchmarks, including ALFWorld (Shridhar et al., 2021) and WebShop (Yao et al., 2022a). ALFWorld is an embodied environment designed to evaluate the ability of an agent to perform long-horizon, multi-step decision-making. The benchmark contains 3,827 task instances spanning six categories of common household activities. WebShop is a large-scale, web-based interactive environment that tests agents in realistic online shopping scenarios. The environment includes over 1.1 million products and approximately 12,000 user instructions. In addition to text-based agentic environments for LLMs, we also consider vision-language model (VLM) settings in interactive game environments: Sokoban benchmark (Schrader, 2018), where agents must reason over visual observations and plan multi-step actions. Detailed descriptions of benchmark you can find in Appendix C.

**Compared Methods.** For LLMs, we compare GraphGPO with several competitive baselines, including: 1) Closed-source LLMs: GPT-4o (Achiam et al., 2024) and Gemini-2.5-Pro (Gemini et al., 2023), which are widely adopted advanced models in general-purpose settings. 2) Prompting-based agents: ReAct (Yao et al., 2022b) and Reflexion (Shinn et al., 2023), which rely entirely on in-context information to accomplish multi-turn tasks without parameter updates. 3) RL-based training methods: PPO (Schulman et al., 2017), a widely used critic-based RL algorithm, as well as group-based RL methods, including RLOO (Kool et al., 2019; Ahmadian et al., 2024), GRPO (Shao et al., 2024), and GiGPO (Feng et al., 2025b). For VLMs, we focus on comparisons with group-based RL methods.

**Implementation Details.** Following prior work (Feng et al., 2025b), we use Qwen2.5-1.5B-Instruct and Qwen2.5-7B-Instruct (Qwen, 2025) as the base LLMs, and Qwen2.5-VL-3B-Instruct (Bai et al., 2025) as the base VLM. For consistency with existing agent frameworks, the agent retains only the most recent two interaction steps as memory and discards earlier history. In addition, the agent is prompted to first generate its reasoning enclosed within <think></think>tags, followed by the action enclosed within <action></action>tags (Wei et al., 2022). For all methods, we adopt the same training hyperparameters to ensure fair comparisons. For group-based methods, the rollout group size is set to 8 across all benchmarks. Moreover, the balancing factors $\beta^G$ and $\beta^E$ are both set to 1 for GraphGPO. For simplicity, we consider all transition costs are unitary, i.e., $c(s, \boldsymbol{a}) = 1$. Detailed implementation details, including training settings and hyperparameters, are provided in the Appendix C.

### 5.2. Experimental Results

**Performance on Agentic Benchmarks.** Table 1 presents the performance comparison between GraphGPO and several competitive baselines on the ALFWorld and WebShop

*Table 1.* The test performance on ALFWorld and WebShop. For ALFWorld, we report the average success rate (%) for each subtask as well as the overall result. For WebShop, we report the average task score and the average success rate (%). Most results are averaged over 3 random seeds during testing. The best performances are highlighted in bold.

| Type | Method | ALFWorld | | | | | | | WebShop | |
|---|---|---|---|---|---|---|---|---|---|---|
| | | Pick | Clean | Cool | Look | Heat | Pick2 | All | Score | Succ. |
| ***Closed-Source Models*** | | | | | | | | | | |
| Prompting | GPT-4o | 75.3 | 60.8 | 31.2 | 56.7 | 21.6 | 49.8 | 48.0 | 31.8 | 23.7 |
| Prompting | Gemini-2.5-Pro | 92.8 | 63.3 | 62.1 | 69.0 | 26.6 | 58.7 | 60.3 | 42.5 | 35.9 |
| ***Qwen2.5-1.5B-Instruct*** | | | | | | | | | | |
| Prompting | Qwen2.5 | 5.9 | 5.5 | 3.3 | 9.7 | 4.2 | 0.0 | 4.1 | 23.1 | 5.2 |
| Prompting | ReAct | 17.4 | 20.5 | 15.7 | 6.2 | 7.7 | 2.0 | 12.8 | 40.1 | 11.3 |
| Prompting | Reflexion | 35.3 | 22.2 | 21.7 | 13.6 | 19.4 | 3.7 | 21.8 | 55.8 | 21.9 |
| RL Training | PPO | $64.8_{(3.5)}$ | $40.5_{(6.9)}$ | $57.1_{(4.9)}$ | $60.6_{(6.6)}$ | $46.4_{(4.0)}$ | $47.4_{(1.9)}$ | $54.4_{(3.1)}$ | $73.8_{(3.0)}$ | $51.5_{(2.9)}$ |
| RL Training | RLOO | $88.3_{(3.0)}$ | $52.8_{(8.6)}$ | $71.0_{(5.9)}$ | $62.8_{(8.7)}$ | $66.4_{(5.5)}$ | $56.9_{(4.7)}$ | $69.7_{(2.5)}$ | $73.9_{(5.6)}$ | $52.1_{(6.7)}$ |
| RL Training | GRPO | $82.89_{(3.62)}$ | $82.14_{(6.37)}$ | $73.86_{(6.84)}$ | $78.57_{(0.00)}$ | $77.78_{(4.54)}$ | $71.43_{(3.89)}$ | $77.86_{(1.33)}$ | $84.73_{(0.49)}$ | $71.35_{(2.05)}$ |
| RL Training | GiGPO | $\mathbf{98.81}_{(1.68)}$ | $95.16_{(3.89)}$ | $81.46_{(0.56)}$ | $78.57_{(0.00)}$ | $94.44_{(0.00)}$ | $\mathbf{93.65}_{(5.94)}$ | $90.88_{(0.97)}$ | $87.94_{(0.43)}$ | $73.83_{(2.30)}$ |
| **RL Training** | **GraphGPO** | $95.15_{(1.62)}$ | $\mathbf{100.0}_{(0.00)}$ | $\mathbf{85.26}_{(2.58)}$ | $\mathbf{85.71}_{(5.83)}$ | $\mathbf{96.30}_{(2.61)}$ | $\mathbf{93.65}_{(2.24)}$ | $\mathbf{92.71}_{(1.32)}$ | $\mathbf{89.29}_{(1.48)}$ | $\mathbf{78.65}_{(3.86)}$ |
| ***Qwen2.5-7B-Instruct*** | | | | | | | | | | |
| Prompting | Qwen2.5 | 33.4 | 21.6 | 19.3 | 6.9 | 2.8 | 3.2 | 14.8 | 26.4 | 7.8 |
| Prompting | ReAct | 48.5 | 35.4 | 34.3 | 13.2 | 18.2 | 17.6 | 31.2 | 46.2 | 19.5 |
| Prompting | Reflexion | 62.0 | 41.6 | 44.9 | 30.9 | 36.3 | 23.8 | 42.7 | 58.1 | 28.8 |
| RL Training | PPO | $92.3_{(4.0)}$ | $64.0_{(8.4)}$ | $92.5_{(2.4)}$ | $89.5_{(7.0)}$ | $80.3_{(2.0)}$ | $68.8_{(8.3)}$ | $80.4_{(2.7)}$ | $81.4_{(3.1)}$ | $68.7_{(5.1)}$ |
| RL Training | RLOO | $87.6_{(4.3)}$ | $78.2_{(8.3)}$ | $87.3_{(5.8)}$ | $81.3_{(7.6)}$ | $71.9_{(5.2)}$ | $48.9_{(8.4)}$ | $75.5_{(4.6)}$ | $80.3_{(3.2)}$ | $65.7_{(4.0)}$ |
| RL Training | GRPO | $88.98_{(5.30)}$ | $91.98_{(4.43)}$ | $77.89_{(4.58)}$ | $78.57_{(0.00)}$ | $90.74_{(5.24)}$ | $71.43_{(3.89)}$ | $83.33_{(2.05)}$ | $84.31_{(1.27)}$ | $75.00_{(2.78)}$ |
| RL Training | GiGPO | $97.53_{(1.75)}$ | $\mathbf{100.0}_{(0.00)}$ | $83.98_{(1.32)}$ | $90.48_{(6.73)}$ | $\mathbf{94.44}_{(0.00)}$ | $\mathbf{100.0}_{(0.00)}$ | $94.27_{(1.33)}$ | $86.72_{(1.44)}$ | $78.38_{(1.94)}$ |
| **RL Training** | **GraphGPO** | $\mathbf{100.0}_{(0.00)}$ | $\mathbf{100.0}_{(0.00)}$ | $\mathbf{91.40}_{(1.50)}$ | $\mathbf{92.86}_{(5.83)}$ | $\mathbf{94.44}_{(0.00)}$ | $92.06_{(2.24)}$ | $\mathbf{95.31}_{(1.10)}$ | $\mathbf{86.94}_{(0.68)}$ | $\mathbf{80.31}_{(1.33)}$ |

*Table 2.* The test performance of VLM agents using Qwen2.5-VL-3B-Instruct on the interactive game environment SokoBan. We report the average task score and the average success rate (%).

| Type | Method | Sokoban [6×6] |
|---|---|---|
| Prompting | Qwen2.5-VL | 11.7 |
| RL Training | GRPO | 67.1 |
| RL Training | GiGPO | 76.92 |
| **RL Training** | **GraphGPO** | $\mathbf{86.98}_{(0.73)}$ |

benchmarks. As shown in the table, GraphGPO consistently outperforms all baselines across both benchmarks. In particular, on ALFWorld, GraphGPO achieves improvements on nearly all subtasks, resulting in average success rate gains of 14.85% and 11.98% over GRPO for the 1.5B and 7B models, respectively. On WebShop, GraphGPO not only attains higher task scores but also improves the average success rate over GRPO by 7.30% and 5.31% for the 1.5B and 7B models, respectively. Furthermore, Table 2 reports the comparison results of GraphGPO with group-based RL baselines in the interactive game environment Sokoban. In this deterministic maze-based setting, GraphGPO outperforms GRPO by 19.88% and surpasses GiGPO by 10.06%. These results validate the effectiveness of GraphGPO in improving agent performance.

**Training Dynamics.** Figure 4 illustrates the evolution of the success rate over training steps on ALFWorld, Web-

Shop, and Sokoban, comparing GraphGPO with GRPO and GiGPO. From Figure 4, we have the following observations: 1) While all three methods exhibit a clear upward trend during training, GraphGPO and GiGPO consistently outperform GRPO across all benchmarks. This suggests that relying solely on coarse-grained episode-level signals is not effective enough for multi-turn agentic tasks. 2) GraphGPO maintains a leading position throughout the training process and achieves the highest peak performance on all benchmarks. This improvement can be attributed to finer-grained credit assignment enabled by graph-based advantage estimation, which leverages global information aggregated from the rollout data. 3) GraphGPO shows a substantially faster improvement in the early stages of training, and the performance gap between GraphGPO and the baselines is most pronounced during the mid-training phase. This behavior indicates that the state-transition graph allows more effective more informative step-level signals, particularly when the overall rollout success rate is low. The training dynamic of validation you can find in Appendix D.2.

**Ablation Study.** Table 3 presents ablation results comparing GraphGPO with GiGPO. First, we evaluate variants without episode-level advantages ($-A^E$), where only step-level advantages ($A^G$ or $A^S$) are used. Under this setting, both GraphGPO and GiGPO exhibit performance degradation across all benchmarks. This indicates that relying

*Table 3.* Ablation study comparing variants without episode-level advantages ($-A^E$) and with dynamic sampling ($+DS$). The best performance under the same setting is highlighted in bold. For ALFWorld and WebShop, agents use Qwen2.5-1.5B-Instruct as the base LLM. We report the average success rate (%) for all settings.

| Method | ALFWorld | | | | | | | WebShop | Sokoban |
| --- | --- | --- | --- | --- | --- | --- | --- | --- | --- |
| | Pick | Clean | Cool | Look | Heat | Pick2 | All | | |
| GiGPO | **98.81**$_{(1.68)}$ | 95.16$_{(3.89)}$ | 81.46$_{(0.56)}$ | 78.57$_{(0.00)}$ | 94.44$_{(0.00)}$ | **93.65**$_{(5.94)}$ | 90.88$_{(0.97)}$ | 73.83$_{(2.30)}$ | 76.92 |
| - $A^E$ | 96.34$_{(0.06)}$ | **100.0**$_{(0.00)}$ | 83.98$_{(1.32)}$ | 83.33$_{(3.36)}$ | **98.15**$_{(2.61)}$ | 85.71$_{(7.78)}$ | 91.41$_{(2.30)}$ | 73.18$_{(2.08)}$ | 62.5 |
| + DS | 97.57$_{(1.72)}$ | **100.0**$_{(0.00)}$ | 90.21$_{(4.53)}$ | 95.24$_{(6.73)}$ | 98.00$_{(2.23)}$ | 98.41$_{(2.24)}$ | 96.35$_{(1.95)}$ | 81.25$_{(2.78)}$ | - |
| GrpahGPO | 95.15$_{(1.62)}$ | **100.0**$_{(0.00)}$ | 85.26$_{(2.58)}$ | **85.71**$_{(5.83)}$ | 96.30$_{(2.61)}$ | **93.65**$_{(2.24)}$ | 92.71$_{(1.32)}$ | **78.65**$_{(3.86)}$ | **86.98**$_{(0.73)}$ |
| - $A^E$ | **97.57**$_{(1.71)}$ | **100.0**$_{(0.00)}$ | 85.22$_{(2.78)}$ | **85.71**$_{(0.00)}$ | 94.44$_{(0.00)}$ | 85.71$_{(0.00)}$ | 91.67$_{(0.97)}$ | 75.00$_{(2.76)}$ | 83.07$_{(2.41)}$ |
| + DS | **100.0**$_{(0.00)}$ | 98.41$_{(2.24)}$ | 95.15$_{(3.43)}$ | 97.62$_{(3.36)}$ | **100.0**$_{(0.00)}$ | **100.0**$_{(0.00)}$ | 98.43$_{(1.28)}$ | 85.68$_{(3.52)}$ | 90.36$_{(1.60)}$ |

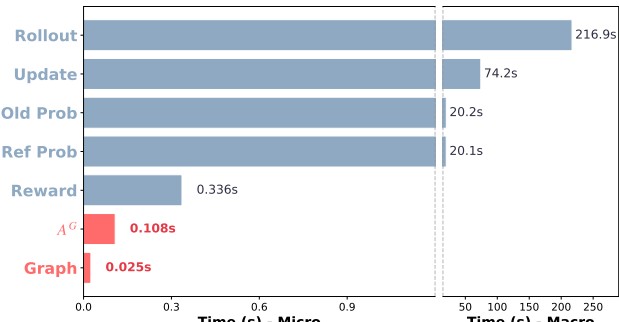

*Figure 5.* Per-iteration runtime breakdown of training stages, including rollout, graph aggregation, reward estimation, graph-based advantage computation, recomputation of old and reference policy probabilities, and policy update. Gray bars denote stages shared by group-based methods, while red bars indicate the additional overhead introduced by GraphGPO. A broken x-axis is used to accommodate stages with smaller runtimes.

solely on step-level advantages is insufficient to provide informative credit signals for all steps in multi-turn agentic tasks. Nevertheless, GraphGPO consistently outperforms GiGPO across all tasks, achieving a notably larger margin on Sokoban, where it surpasses GiGPO by 20.57%. Second, we compare GraphGPO with GiGPO under dynamic sampling (Yu et al., 2025b). For each example $x$, a set of rollout trajectories $\{\tau_1, \tau_2, \ldots, \tau_M\}$ is resampled if all trajectories in the set are either successful or failed. This resampling process continues until a sufficient number of trajectories is collected or a maximum of 10 attempts is reached. Under this setting, both GraphGPO and GiGPO benefit from dynamic sampling. However, GraphGPO consistently achieves superior performance.

**Computational Overhead.** For memory overhead, GraphGPO constructs states using only the deterministic components of environment observations (see Appendix for details), without storing additional models or auxiliary datasets. During training, the state-transition graph is maintained using a hash table whose size is $|\mathcal{E}|$, which scales with the rollout parameters $M$ and $T$ and remains

negligible in practice. Regarding per-iteration time overhead, GraphGPO performs a single shortest-path search using Dijkstra algorithm (Haeupler et al., 2024), with a time complexity of $O((|\mathcal{V}| + |\mathcal{E}|) \log |\mathcal{V}|)$. Compared with the computational cost of updating LLMs, this overhead is negligible. The per-iteration runtime breakdown is shown in Figure 5. As illustrated, the most time-consuming stage in training is rollout, which samples trajectories from the policy to guide optimization, followed by policy updates involving backpropagation. In comparison, the additional costs introduced by GraphGPO mainly come from graph construction (0.108s) and graph-based advantage computation (0.025s). These costs are negligible when compared with rollout (216.9s) and policy update (74.2s), accounting for only 0.04% of the total per-iteration runtime. Since rollout is the most time-consuming component, improving RL training efficiency crucially depends on recovering high-fidelity step-level supervision from collected rollouts, especially when successes are sparse.

Additional experimental results, including training dynamics of validation, hyperparameter studies and case study , are provided in Appendix D.

## 6. Conclusion

In this work, we introduced Graph-based Group Policy Optimization (GraphGPO), a simple and effective method for enabling faithful step-level credit assignment in multi-turn agentic tasks. Unlike existing methods that rely on trajectory-level attribution according to final outcomes, GraphGPO aggregates rollout trajectories into a unified state-transition graph and assigns step-level credit based on global progress toward the task goal. Overall, extensive experiments across diverse multi-turn agentic benchmarks demonstrate that GraphGPO consistently outperforms prior group-based methods by exploiting global structural information across rollouts for step-level credit assignment. The proposed formulation complements existing group-based RL methods and provides a practical mechanism for improving credit assignment in long-horizon agentic tasks.

## Impact Statement

This paper presents work whose goal is to advance the field of Machine Learning. There are many potential societal consequences of our work, none which we feel must be specifically highlighted here.

## Acknowledgements

This research is supported by the RIE2025 Industry Alignment Fund – Industry Collaboration Projects (IAF-ICP) (Award I2301E0026), administered by A*STAR, as well as supported by Alibaba Group and NTU Singapore through Alibaba-NTU Global e-Sustainability CorpLab (ANGEL).

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

# A. Pseudo Code.

---

**Algorithm 1** The pseudo-code of GraphGPO

---

1: **Require:** Initial policy $\pi_{\theta_{\text{old}}}$, task distribution $p(X)$, distance discount factor $\omega$, clipping parameter $\epsilon$, KL penalty $\beta$, group size $M$, maximum step $T$
2: **for** each training iteration **do**
3:    Update the old policy model: $\theta_{\text{old}} \leftarrow \theta$
4:    // Multi-step rollout phase
5:    Sample task $\boldsymbol{x} \sim p(X)$ and initialize $M$ identical environments
6:    **for** $t = 1$ to $T$ **do**
7:      Sample actions $\left\{ \boldsymbol{a}_t^m \sim \pi_{\theta_{\text{old}}}(\cdot \mid \boldsymbol{s}_t^m, \boldsymbol{x}) \right\}_{m=1}^{M}$
8:      Execute actions, observe the next state $\{ \boldsymbol{s}_{t+1}^m \}_{m=1}^{M}$
9:    **end for**
10:   // Graph aggregation phase
11:   *Construct directed graphs $\mathcal{G} = (\mathcal{S}, \mathcal{E})$ by Section (4.2)*
12:   *Compute distance $d(s)$ for each nodes $s \in \mathcal{S}$ by Eq. (3)*
13:   // Grouping phase
14:   *Build step-level groups $G^G(s)$ by Eq. (5)*
15:   // Advantage computation phase
16:   *Compute episode-level advantages by Eq. (2)*
17:   *Compute graph-based advantages within each group by Eq. (6)*
18:   // Policy update phase
19:   Update policy $\theta$ by maximizing objective Eq. (9)
20: **end for**

---

Algorithm 1 summarizes the full training procedure. We highlight the key stages of the training pipeline, with the runtime of each stage reported in Figure 5. As such, GraphGPO preserves the critic-free, low-memory, and stable convergence properties, as well as the high efficiency of group-based RL, while aggregating all rollout trajectories into a global structure to capture latent relationships across steps, thereby enabling more fine-grained and informative credit assignment.

# B. Proofs of Theorem

## B.1. Proof of Proposition 4.1

By definition of the graph-based reward, we have graph-based step- level rewards for edge $(s, \boldsymbol{a}_{\text{good}}, s'_{\text{good}})$ and $(s, \boldsymbol{a}_{\text{bad}}, s'_{\text{bad}})$ as follows:

$$R^G(s, \boldsymbol{a}_{\text{good}}, s'_{\text{good}}) = r_{\text{succ}} \, \omega^{d(s'_{\text{good}}) + c(s, \boldsymbol{a}_{\text{good}})}, \tag{10}$$

$$R^G(s, \boldsymbol{a}_{\text{bad}}, s'_{\text{bad}}) = r_{\text{succ}} \, \omega^{d(s'_{\text{bad}}) + c(s, \boldsymbol{a}_{\text{bad}})}. \tag{11}$$

where $r_{\text{succ}} > 0$ is a scalar. let $D_{\text{good}} = d(s'_{\text{good}}) + c(s, \boldsymbol{a}_{\text{good}})$ and $D_{\text{bad}} = d(s'_{\text{bad}}) + c(s, \boldsymbol{a}_{\text{bad}})$. By the assumption that $\boldsymbol{a}_{\text{good}}$ moves closer to the goal than $\boldsymbol{a}_{\text{bad}}$, we have:

$$D_{\text{good}} < D_{\text{bad}}. \tag{12}$$

Since $\omega \in (0, 1)$, the function $x \rightarrow \omega^x$ is strictly decreasing on $\mathcal{R}$, so we have:

$$D_{\text{good}} < D_{\text{bad}} \implies \omega^{D_{\text{good}}} > \omega^{D_{\text{bad}}} \implies R^G(s, \boldsymbol{a}_{\text{good}}, s'_{\text{good}}) > R^G(s, \boldsymbol{a}_{\text{bad}}, s'_{\text{bad}}). \tag{13}$$

Then we recall the graph-based advantages $A^G(s, \boldsymbol{a}_{\text{good}}, s'_{\text{good}}) = \frac{R^G(s, \boldsymbol{a}_{\text{good}}, s'_{\text{good}}) - \mu(G^G(s))}{\sigma(G^G(s))}$ and $A^G(s, \boldsymbol{a}_{\text{bad}}, s'_{\text{bad}}) = \frac{R^G(s, \boldsymbol{a}_{\text{bad}}, s'_{\text{bad}}) - \mu(G^G(s))}{\sigma(G^G(s))}$ where $\mu(G^G(s))$ and $\sigma(G^G(s))$ are the same for all actions at state $s$. Since there are at least the two edges considered above and their rewards are different, we have $\sigma(G^G(s)) > 0$. Hence,

$$A^G(s, \boldsymbol{a}_{\text{good}}, s'_{\text{good}}) - A^G(s, \boldsymbol{a}_{\text{bad}}, s'_{\text{bad}}) = \frac{R^G(s, \boldsymbol{a}_{\text{good}}, s'_{\text{good}}) - \mu(G^G(s))}{\sigma(G^G(s))} - \frac{R^G(s, \boldsymbol{a}_{\text{bad}}, s'_{\text{bad}}) - \mu(G^G(s))}{\sigma(G^G(s))} \tag{14}$$

$$= \frac{R^G(s, \boldsymbol{a}_{\text{bad}}, s'_{\text{bad}}) - R^G(s, \boldsymbol{a}_{\text{bad}}, s'_{\text{bad}})}{\sigma(G^G(s))} \tag{15}$$

$$> 0 \tag{16}$$

Which concludes the proof of Proposition 4.1. □

### B.2. Proof of Proposition 4.2

We analyze the conditional variance in each step-level feedback.

For a state–action pair $(s, \boldsymbol{a}, s')$ that is visited with non-zero probability, the trajectory-style step feedback at time $t$ as follows:

$$X^S = \eta(t)R(\boldsymbol{\tau}), \tag{17}$$

where $\eta(t) = 1$ for GRPO and $\eta(t) = \lambda^{T-t}$ for GRPO. The graph-style step feedback at time $t$ as follows:

$$X^G = R^G(s, \boldsymbol{a}, s'), \tag{18}$$

Suppose positive probability of both successful and failed trajectories passing through $(s, \boldsymbol{a}, s')$. Denote by $\tau_{succ}(s, \boldsymbol{a}, s')$ and $\tau_{fail}(s, \boldsymbol{a}, s')$ the sets of successful and failed trajectories that visit $(s, \boldsymbol{a}, s')$ with positive probability, i.e.,

$$p_{succ} = \mathbb{P}(\tau \in \tau_{succ}(s, a)), \tag{19}$$
$$p_{fail} = 1 - p_{succ} > 0. \tag{20}$$

Suppose $(s, \boldsymbol{a}, s')$ is independent with step $t$, that is, $p(s, \boldsymbol{a}, s'|t) = p(s, \boldsymbol{a}, s')$. Then, for a $\tau \in \tau_{succ}(s, \boldsymbol{a}, s')$ we have $R(\tau) > 0$, so:

$$X^S = \eta(t)R(\boldsymbol{\tau}) > 0. \tag{21}$$

It is worth noting that $X^S$ is not independent with step $t$. $X^S$ is different when $(s, \boldsymbol{a}, s')$ observed in different $t$, that is:

$$Var(X^S|s, \boldsymbol{a}, s', \tau_{succ}) \geq 0. \tag{22}$$

For a fail trajectory $\tau \in \tau_{fail}(s, \boldsymbol{a}, s')$, the reward $R(\tau) = 0$, so $X^S = 0$ and $Var(X^S|s, \boldsymbol{a}, s', \tau_{\text{fail}}) = 0$. Combining the above Eq. 22, we have the final conditional variance for he trajectory-style step feedback:

$$Var(X^S|s, \boldsymbol{a}, s') \geq 0. \tag{23}$$

Now consider the graph-style step feedback. The empirical state-transition graph and distance function $d(\cdot)$ are constructed from the current batch of rollouts. Once the graph is fixed, the reward assigned to any observed transition $(s, \boldsymbol{a}, s')$ is a deterministic function $R^G(s, \boldsymbol{a}, s')$. Since the environment is deterministic, the empirical graph is deterministic. The successor state $s'$ and thus the feedback $X^G$ are deterministic. Therefore

$$Var(X^G|s, \boldsymbol{a}, s') = 0. \tag{24}$$

Combining the Eq. 23, we have:

$$Var(X^G|s, \boldsymbol{a}, s') \leq Var(X^S|s, \boldsymbol{a}, s'). \tag{25}$$

Which concludes the proof of Proposition 4.2. □

## C. Implementation Details

### C.1. Computing Details

All experiments are conducted for 150 training epochs across all benchmarks and model configurations. For experiments using Qwen2.5-1.5B-Instruct and Qwen2.5-VL-3B-Instruct, training is performed on 2 NVIDIA H100 GPUs with 80 GB memory each. For experiments using the larger Qwen2.5-7B-Instruct model, we employ 4 NVIDIA H100 GPUs with 80 GB memory each to accommodate the increased computational and memory requirements.

---

**Prompt Template for ALFWorld**

You are an expert agent operating in the ALFRED embodied Environment. Your task is to: {task_description}. Prior to this step, you have already taken {step_count} step(s). Below are the most recent {history_length} observations and the corresponding actions you took: {action_history}. You are now at step {current_step} and your current observation is: {current_observation}. Your admissible actions of the current situation are: [{admissible_actions}]. Location: {current_location}. Items in hand (status): {current_holding}. History moving list by you: {history_moving}.

Now it's your turn to take an action. You should first reason step-by-step about the current situation. This reasoning process MUST be enclosed within <think> </think> tags. Once you've finished your reasoning, you should choose an admissible action for current step and present it within <action > </action> tags.

---

*Figure 6.* Prompt template used for ALFWorld experiments.

---

**Prompt Template for WebShop**

You are an expert autonomous agent operating in the WebShop e-commerce environment. Your task is to: {task_description}. Prior to this step, you have already taken {step_count} step(s). Below are the most recent {history_length} observations and the corresponding actions you took: {action_history}. You are now at step {current_step} and your current observation is: {current_observation}. Your admissible actions for the current situation are: [{admissible_actions}].

Now it's your turn to take one action for the current step. You should first reason step-by-step about the current situation, then think carefully which admissible action best advances the shopping goal. This reasoning process MUST be enclosed within <think> </think> tags. Once you've finished your reasoning, you should choose an admissible action for current step and present it within <action > </action> tags.

---

*Figure 7.* Prompt template used for WebShop experiments.

## C.2. Prompts

The prompt templates used for agents in ALFWorld, WebShop, and Sokoban are shown in Figure 6, Figure 7, and Figure 8, respectively. All prompts are constructed using Python-style string formatting, where placeholders enclosed in curly braces ({}) indicate semantic slots that are dynamically instantiated at each interaction step.

Specifically, {task_description} specifies the task definition, and {step_count} denotes the number of actions already executed. The placeholder {history_length} indicates the length of the visible interaction history, which is set to 2 in all experiments. The agent's recent action–observation history is represented by {action_history}. The current interaction step is denoted by {current_step}, while {current_observation} corresponds to the observation returned by the environment at the current step. The placeholder {admissible_actions} enumerates the set of valid actions available to the agent under the current observation.

In addition, several environment-specific placeholders are introduced to provide richer state information in ALFWorld. {current_location} indicates the agent's current location, {current_holding} describes the object(s) currently held by the agent along with their states, and {history_moving} records the agent's object manipulation history. When no object movement has occurred, the {history_moving} field is omitted from the prompt.

To explicitly structure the model's reasoning and outputs, we employ a set of control tags. The step-by-step reasoning process is enclosed within <think> and </think> tags, while the final selected action is wrapped within <action> and </action> tags. For vision–language settings, <image> serves as a placeholder token representing the visual observation.

## C.3. Comparing Methods

**GPT-4o.** A closed-source, large-scale large language model used as a strong baseline for multi-turn agentic tasks, demonstrating advanced reasoning and instruction-following capabilities.

**Gemini-2.5-Pro.** A closed-source large language model comparable in scale and overall capability to GPT-4o, serving as another competitive proprietary baseline.

**ReAct.** A prompting-based agent framework that interleaves reasoning and acting by explicitly generating intermediate thoughts and actions in a chain-of-thought manner.

---

Prompt Template for Sokoban

You are an expert agent operating in the Sokoban environment. Your goal is to push all the boxes onto the target spots. Once all boxes are on the targets, you win!
# Rules
You can only push boxes. You can't pull them, so plan ahead to avoid getting stuck.
You can't walk through or push boxes into walls.
To avoid traps, do not push boxes into corners or against walls where they can't be moved again.
# Visual Elements in the Image:
Character: A small, green alien-like figure with two antennae and black eyes. It represents you.
Box: A yellow crate marked with an orange "X" across its front. It is the box you need to push.
Target: A black tile outlined in red, with a small red diamond shape in the center. It marks the destination where a box should be pushed.
# Current Step
Your current observation is shown in the image: \<image\>
Your admissible actions are ["up", "down", "left", "right"].
Now it's your turn to make a move (choose ONE action only for the current step).
You should first reason step-by-step about the current situation — observe the positions of boxes and targets, plan a path to push a box toward a target, and avoid traps like corners or walls. This reasoning process MUST be enclosed within \<think\> \</think\> tags. Once you've finished your reasoning, you should choose an admissible action for current step and present it within \<action \> \</action\> tags.

---

*Figure 8.* Prompt template used for Sokoban experiments.

**Reflexion.** A prompting-based agent that enhances performance through self-reflection and iterative refinement over previously generated trajectories.

**PPO.** Proximal Policy Optimization, a widely used actor–critic reinforcement learning algorithm that relies on a learned value function for stable policy updates.

**RLOO.** Reinforcement Learning with Offline Observations, a group-based reinforcement learning approach that estimates advantages using group-level statistics without training an explicit value network.

**GRPO.** Group Relative Policy Optimization, a group-based reinforcement learning method that performs trajectory-level advantage estimation and is designed to scale RL training to multi-step and reasoning-intensive tasks.

**GiGPO.** Grouped Incremental Group Policy Optimization, a hierarchical group-based reinforcement learning method that performs groupwise step-level advantage estimation for LLM-based agents.

For the main group-based baselines (GRPO and GiGPO), we reproduce their experimental results under the same settings. For other comparison methods, we directly adopt the reported results from GiGPO.

### C.4. ALFWorld

All methods are configured with identical hyperparameters for fair comparison. The maximum prompt length is set to 2048 tokens, and the maximum response length is 512 tokens. Each episode allows up to 50 environment steps. The learning rate is set to $1 \times 10^{-6}$ for the actor and $1 \times 10^{-5}$ for the critic, where the critic is used only in PPO. We adopt a rule-based reward scheme, assigning a reward of 10 for successful task completion and 0 otherwise. To handle invalid actions generated by the agent, a reward penalty of $-0.1$ is applied.

For all group-based RL methods, we use a group size of 8 and sample 16 groups per rollout, resulting in a total of $16 \times 8 = 128$ environments. In contrast, PPO uses 128 independent environments for rollouts. The rollout temperature is set to 1.0, while the validation temperature is set to 0.4. The mini-batch size is 256, and the KL-divergence loss coefficient is set to 0.01. For GiGPO, the discount factor $\lambda$ is set to 0.95.

It is worth noting that we identify incomplete observations and issues in the default ALFWorld environment. To address this, we augment the observations with additional information to improve environment fidelity. Specifically, for each environment step, we add only the agent's current location and the objects it is holding. Detailed descriptions of these modifications are provided in Appendix E. For all group-based RL methods, including GRPO and GiGPO, we reproduce their experimental results under this improved environment. Compared to the original setting, these methods consistently achieve noticeably

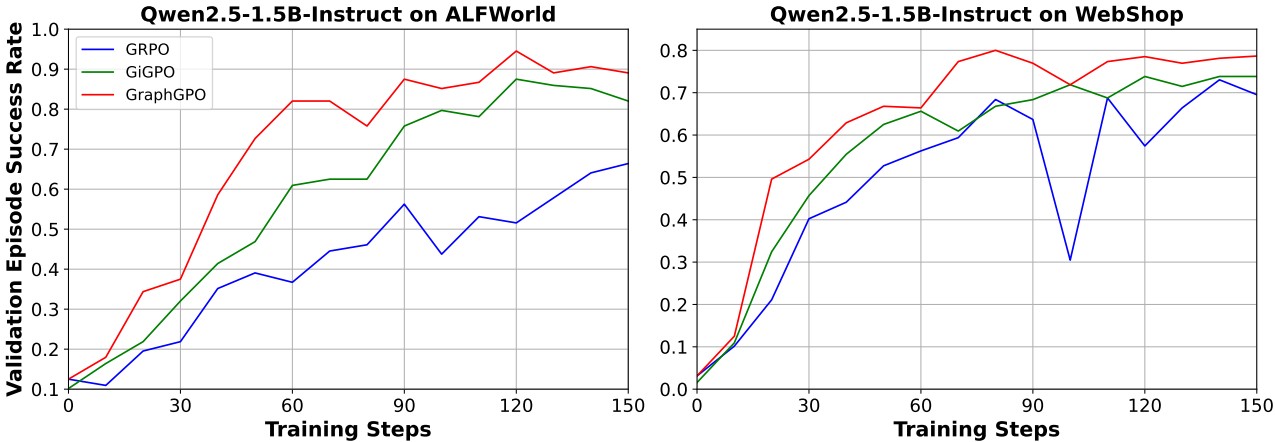

*Figure 9.* Validation episode success rate versus training steps for GraphGPO (red), GiGPO (green), and GRPO (blue) on the ALFWorld and WebShop benchmarks. The success rate is recorded every 10 training steps.

better performance. Finally, the distance discount factor $\omega$ in GraphGPO is set to 0.10.

### C.5. WebShop

All methods are configured with identical hyperparameters to ensure fair comparison. The maximum prompt length is set to 5120 tokens, and the maximum response length is 512 tokens. Each episode is limited to 15 environment steps. The learning rate is $1 \times 10^{-6}$ for the actor and $1 \times 10^{-5}$ for the critic, where the critic is used only in PPO. We adopt a rule-based reward scheme, assigning a reward of 10 for successful task completion and 0 otherwise. Invalid actions are penalized with a reward of $-0.1$.

As with ALFWorld, all group-based RL methods use a group size of 8 and sample 16 groups per rollout, resulting in a total of $16 \times 8 = 128$ environments. In contrast, PPO uses 128 independent environments for rollouts. The rollout temperature is set to 1.0, while the validation temperature is set to 0.4. The mini-batch size is 64, and the KL-divergence loss coefficient is set to 0.01. For GiGPO, the discount factor $\lambda$ is set to 0.95.

It is worth noting that we adopt a more realistic variant of the environment, namely the official *text-rich* version, which provides additional information on interactive elements such as buttons. For all group-based RL methods, including GRPO and GiGPO, we reproduce their experimental results under this enhanced environment. Compared to the original setting, these methods consistently achieve noticeably better performance. Finally, the distance discount factor $\omega$ in GraphGPO is set to 0.20.

### C.6. Sokoban

We evaluate GraphGPO in a vision-based interactive game environment using the Sokoban benchmark. All methods are configured with identical hyperparameters to ensure fair comparison. We use Qwen2.5-VL-3B-Instruct as the base vision language model. The maximum prompt length is set to 1024 tokens, and the maximum response length is 512 tokens. Each episode is limited to 15 environment steps.

The learning rate for the actor is set to $1 \times 10^{-6}$. We adopt a rule-based reward scheme, assigning a reward of 10 for successful task completion and 0 otherwise. Invalid actions generated by the agent are penalized with a reward of $-0.1$. The rollout temperature is set to 1.0, while the validation temperature is set to 0.4. The mini-batch size is 64, and the KL-divergence loss coefficient is set to 0.01.

For group-based reinforcement learning methods, including GRPO and GiGPO, we use a group size of 8, corresponding to 8 parallel rollouts per state. A total of 128 environments are used during training. For GiGPO, the discount factor $\lambda$ is set to 0.95. The distance discount factor $\omega$ in GraphGPO is set to 0.80.

*Table 4.* The test performance on AlfWorld with distance discount $\omega$ changes using the Qwen2.5-1.5B-Instruct. We report the average success rate (%). All results are averaged over 3 random seeds during testing. The best performances are highlighted in bold.

| Task | $\omega = 0.1$ | $\omega = 0.2$ | $\omega = 0.4$ | $\omega = 0.6$ | $\omega = 0.8$ |
|---|---|---|---|---|---|
| Pick | $95.15_{(1.62)}$ | $96.34_{(0.06)}$ | $96.34_{(0.06)}$ | $92.68_{(0.12)}$ | $\mathbf{97.57_{(1.72)}}$ |
| Clean | $\mathbf{100.0_{(0.00)}}$ | $\mathbf{100.0_{(0.00)}}$ | $\mathbf{100.0_{(0.00)}}$ | $\mathbf{100.0_{(0.00)}}$ | $\mathbf{100.0_{(0.00)}}$ |
| Cool | $\mathbf{85.26_{(2.58)}}$ | $83.98_{(1.32)}$ | $82.70_{(1.84)}$ | $83.98_{(1.32)}$ | $83.98_{(1.32)}$ |
| Look | $\mathbf{85.71_{(5.83)}}$ | $\mathbf{85.71_{(0.00)}}$ | $\mathbf{85.71_{(0.00)}}$ | $80.95_{(3.36)}$ | $\mathbf{85.71_{(0.00)}}$ |
| Heat | $96.30_{(2.61)}$ | $92.59_{(2.61)}$ | $\mathbf{100.0_{(0.00)}}$ | $94.44_{(0.00)}$ | $94.44_{(0.00)}$ |
| Pick2 | $\mathbf{93.65_{(2.24)}}$ | $92.06_{(4.48)}$ | $\mathbf{93.65_{(2.24)}}$ | $88.89_{(2.24)}$ | $90.48_{(3.89)}$ |
| All | $92.71_{(1.32)}$ | $91.93_{(0.97)}$ | $\mathbf{92.97_{(0.63)}}$ | $90.36_{(0.36)}$ | $92.19_{(0.64)}$ |

*Table 5.* The test performance on WebShop with distance discount $\omega$ changes using the Qwen2.5-1.5B-Instruct. We report the average success rate (%). All results are averaged over 3 random seeds during testing. The best performances are highlighted in bold.

| Metric | $\omega = 0.2$ | $\omega = 0.4$ | $\omega = 0.6$ | $\omega = 0.8$ | $\omega = 0.95$ |
|---|---|---|---|---|---|
| Succ. | $\mathbf{78.65_{(3.86)}}$ | $75.78_{(3.76)}$ | $78.00_{(2.41)}$ | $75.00_{(2.23)}$ | $75.91_{(4.35)}$ |

# D. Additional Results

### D.1. Hyperparameter

GraphGPO introduces only one method-specific hyperparameter, the distance discount factor $\omega$, while all other hyperparameters follow standard group-based RL settings. Table 4 reports the average success rate on ALFWorld under different values of $\omega$, using Qwen2.5-1.5B-Instruct as the base model. As shown in the table, although varying $\omega$ has some influence on performance, the overall sensitivity is relatively mild. Even in the worst-case setting, GraphGPO achieves performance comparable to the baseline, while in most cases it consistently outperforms the baseline.

A similar trend is observed on WebShop. Table 5 presents the average success rate on WebShop under different values of the distance discount factor $\omega$, again using Qwen2.5-1.5B-Instruct as the base model. Overall, these results indicate that GraphGPO is robust to the choice of $\omega$. In practice, we recommend using a relatively small value of $\omega$, such as in the range of 0.1 to 0.4.

### D.2. Training Dynamics

Figure 4 illustrates the evolution of the training episode success rate over training steps. As shown in Figure 4, GraphGPO consistently outperforms GRPO and GiGPO throughout the entire training process, with particularly pronounced advantages in the early and mid training stages. This behavior can be attributed to the more accurate and informative step-level credit assignment provided by GraphGPO, which enables the policy to receive meaningful learning signals even when overall trajectory success is sparse.

Figure 9 reports the evolution of the validation episode success rate over training steps on ALFWorld and WebShop. Similar trends to the training curves can be clearly observed: GraphGPO maintains superior performance compared to GRPO and GiGPO across the whole training horizon. Moreover, GraphGPO exhibits faster convergence and higher training efficiency. On ALFWorld, GraphGPO reaches comparable validation performance approximately 30 training steps earlier than GiGPO and about 110 steps earlier than GRPO. On WebShop, GraphGPO achieves similar performance roughly 80 steps earlier than GiGPO and 90 steps earlier than GRPO.

Overall, these results indicate that GraphGPO is able to more effectively exploit the information contained in rollout trajectories by leveraging global structural relationships across states. By providing faithful and low-variance step-level supervision beyond trajectory-level outcomes, GraphGPO accelerates policy improvement and leads to faster and more stable convergence in long-horizon multi-turn agentic tasks.

### D.3. Noise Environments

In high-dimensional or long-text observations, minor variations in formatting or environment responses can indeed lead to a fragmented graph. Considering a extreme case where no states can be matched, GraphGPO relies only on the trajectory-level

*Table 6.* The test performance under different noise settings on WebShop. We compare raw-text matching and embedding-based matching, where $p$ denotes the probability of inserting advertisements into a webpage, and $q$ denotes the number of inserted advertisements. We report the average success rate (%). All results are averaged over 3 random seeds during testing. The best performances are highlighted in bold.

| Noise setting | Raw-text matching | Embedding-based matching |
|---|---|---|
| $(q = 1, p = 0.3)$ | $77.85_{(3.15)}$ | $\mathbf{78.26}_{(2.58)}$ |
| $(q = 1, p = 0.6)$ | $75.27_{(2.15)}$ | $\mathbf{77.42}_{(2.38)}$ |
| $(q = 2, p = 0.3)$ | $76.85_{(3.60)}$ | $\mathbf{78.22}_{(1.30)}$ |
| $(q = 2, p = 0.6)$ | $74.22_{(2.60)}$ | $\mathbf{78.26}_{(2.58)}$ |

*Table 7.* The test performance under different group sizes on WebShop. We compare GRPO, GiGPO, and GraphGPO. We report the average success rate (%). All results are averaged over 3 random seeds during testing. The best performances are highlighted in bold.

| Group size | GRPO | GiGPO | GraphGPO |
|---|---|---|---|
| $n = 4$ | $65.23_{(1.99)}$ | $69.90_{(2.19)}$ | $\mathbf{74.06}_{(2.88)}$ |
| $n = 6$ | $66.41_{(2.78)}$ | $73.34_{(1.98)}$ | $\mathbf{74.58}_{(3.00)}$ |
| $n = 8$ | $71.35_{(2.05)}$ | $73.83_{(2.30)}$ | $\mathbf{78.65}_{(3.86)}$ |

advantage and degenerates to GRPO. Therefore, GRPO can be viewed as a lower bound of GraphGPO in the extreme case.

To further examine the applicability of GraphGPO in such settings, we constructed a more challenging version of WebShop by inserting random advertisements generated by Claude. These injected advertisements act as random noise, making exact raw-text matching much more difficult and resulting in a highly sparse graph. To address this issue, we additionally used embedding-based state matching: two states are treated as identical when their embedding similarity exceeds 95%. Table 6 reports experimental results. With embedding-based matching, GraphGPO remains highly robust even under severe graph sparsity and substantial observation noise.

### D.4. Group Sizes

In general, more exploration can better reflect the environment structure, leading to more accurate credit assignment and potentially better performance. However, GraphGPO does not require a complete state-transition graph in order to work effectively. Constructing a fully explored graph would indeed be computationally expensive, especially since rollout is the most time-consuming part of each training iteration, as shown in Figure 5. To examine this trade-off, we provide additional results under different numbers of rollouts in Table 7.

## E. Case Study

Follow, we present a complete case study of an agent trained by GroupGPO on ALFWorld, including the prompt and the response at each step. For each prompt, the components used to define the state in GroupGPO are highlighted in yellow.

---

**Prompt (ALFWorld, Step 1)**

You are an expert agent operating in the ALFRED Embodied Environment.
Your current observation is: -= Welcome to TextWorld, ALFRED! =-You are in the middle of a room. Looking quickly around you, you see a countertop 1, a drawer 6, a drawer 5, a drawer 4, a drawer 3, a drawer 2, a drawer 1, a garbagecan 1, a handtowelholder 2, a handtowelholder 1, a sinkbasin 1, a toilet 1, a toiletpaperhanger 1, a towelholder 2, and a towelholder 1.
Your task is to: clean some cloth and put it in countertop. Location: middle of a room. Items in hand (status): nothing.
Your admissible actions of the current situation are: [go to countertop 1, go to drawer 1, go to drawer 2, go to drawer 3, go to drawer 4, go to drawer 5, go to drawer 6, go to garbagecan 1, go to handtowelholder 1, go to handtowelholder 2, go to sinkbasin 1, go to toilet 1, go to toiletpaperhanger 1, go to towelholder 1, go to towelholder 2, help, inventory, look].
Now it's your turn to take an action.\nYou should first reason step-by-step about the current situation. This reasoning process MUST be enclosed within <think> </think> tags. \n Once you've finished your reasoning, you should choose an admissible action for current step and present it within <action> </action> tags.

---

**Response (ALFWorld, Step 1)**

<think> The task is to clean some cloth and put it on the countertop. We need to consider the actions we can take to achieve this.
1. First, we need to find a cloth.
2. Then, we can clean the cloth using the provided supplies.
3. Finally, we should put the cleaned cloth on the countertop. </think>
<action>go to countertop 1</action>

---

**Prompt (ALFWorld, Step 2)**

You are an expert agent operating in the ALFRED Embodied Environment.
Your task is to: clean some cloth and put it in countertop.
Prior to this step, you have already taken 1 step(s). Below are the most recent 1 observaitons and the corresponding actions you took: [Observation 1: '-= Welcome to TextWorld, ALFRED! =-You are in the middle of a room. Looking quickly around you, you see a countertop 1, a drawer 6, a drawer 5, a drawer 4, a drawer 3, a drawer 2, a drawer 1, a garbagecan 1, a handtowelholder 2, a handtowelholder 1, a sinkbasin 1, a toilet 1, a toiletpaperhanger 1, a towelholder 2, and a towelholder 1.
Your task is to: clean some cloth and put it in countertop. Location: middle of a room. Items in hand (status): nothing.',
Action 1: 'go to countertop 1']
You are now at step 2 and your current observation is: You arrive at countertop 1. On the countertop 1, you see a cloth 2, and a sink 1. Location: countertop 1. Items in hand (status): nothing.
Your admissible actions of the current situation are: [examine countertop 1, go to drawer 1, go to drawer 2, go to drawer 3, go to drawer 4, go to drawer 5, go to drawer 6, go to garbagecan 1, go to handtowelholder 1, go to handtowelholder 2, go to sinkbasin 1, go to toilet 1, go to toiletpaperhanger 1, go to towelholder 1, go to towelholder 2, help, inventory, look, take cloth 2 from countertop 1].
Now it's your turn to take an action.\nYou should first reason step-by-step about the current situation. This reasoning process MUST be enclosed within <think> </think> tags. \n Once you've finished your reasoning, you should choose an admissible action for current step and present it within <action> </action> tags.

---

**Response (ALFWorld, Step 2)**

<think>The current situation is that you are at the countertop 1, where you see a cloth 2 and a sink 1. You have already decided to clean the cloth 2.</think>
<action>take cloth 2 from countertop 1</action>

---

**Prompt (ALFWorld, Step 3)**

You are an expert agent operating in the ALFRED Embodied Environment.
Your task is to: clean some cloth and put it in countertop.
Prior to this step, you have already taken 2 step(s). Below are the most recent 2 observaitons and the corresponding actions you took: [Observation 1: '-= Welcome to TextWorld, ALFRED! =-You are in the middle of a room. Looking quickly around you, you see a countertop 1, a drawer 6, a drawer 5, a drawer 4, a drawer 3, a drawer 2, a drawer 1, a garbagecan 1, a handtowelholder 2, a handtowelholder 1, a sinkbasin 1, a toilet 1, a toiletpaperhanger 1, a towelholder 2, and a towelholder 1.
Your task is to: clean some cloth and put it in countertop. Location: middle of a room. Items in hand (status): nothing.',
Action 1: 'go to countertop 1']
[Observation 2: 'You arrive at countertop 1. On the countertop 1, you see a cloth 2, and a sink 1. Location: countertop 1. Items in hand (status): nothing.', Action 2: 'take cloth 2 from countertop 1']
You are now at step 3 and your current observation is: You pick up the cloth 2 from the countertop 1. Location: countertop 1. Items in hand (status): cloth 2(unprocessed).
Your admissible actions of the current situation are: [examine cloth 2, examine countertop 1, go to drawer 1, go to drawer 2, go to drawer 3, go to drawer 4, go to drawer 5, go to drawer 6, go to garbagecan 1, go to handtowelholder 1, go to handtowelholder 2, go to sinkbasin 1, go to toilet 1, go to toiletpaperhanger 1, go to towelholder 1, go to towelholder 2, help, inventory, look, move cloth 2 to countertop 1].
Now it's your turn to take an action.\nYou should first reason step-by-step about the current situation. This reasoning process MUST be enclosed within <think> </think> tags. \n Once you've finished your reasoning, you should choose an admissible action for current step and present it within <action> </action> tags.

---

**Response (ALFWorld, Step 3)**

<think>The cloth is currently picked up from the countertop 1. The cloth currently is not yet cleaned. We need to clean the cloth before placing it in the countertop.</think>
<action>go to sinkbasin 1</action>

**Prompt (ALFWorld, Step 4)**

You are an expert agent operating in the ALFRED Embodied Environment.

Your task is to: clean some cloth and put it in countertop.

Prior to this step, you have already taken 3 step(s). Below are the most recent 2 observaitons and the corresponding actions you took: [Observation 2: 'You arrive at countertop 1. On the countertop 1, you see a cloth 2, and a sink 1. Location: countertop 1. Items in hand (status): nothing.', Action 2: 'take cloth 2 from countertop 1']

[Observation 3: 'You pick up the cloth 2 from the countertop 1. Location: countertop 1. Items in hand (status): cloth 2(unprocessed).', Action 3: 'go to sinkbasin 1']

You are now at step 4 and your current observation is: You arrive at sinkbasin 1. On the sinkbasin 1, you see a sink 1, and a soapbar 2. Location: sinkbasin 1. Items in hand (status): cloth 2(unprocessed).

Your admissible actions of the current situation are: [clean cloth 2 with sinkbasin 1, examine cloth 2, examine sinkbasin 1, go to countertop 1, go to drawer 1, go to drawer 2, go to drawer 3, go to drawer 4, go to drawer 5, go to drawer 6, go to garbagecan 1, go to handtowelholder 1, go to handtowelholder 2, go to toilet 1, go to toiletpaperhanger 1, go to towelholder 1, go to towelholder 2, help, inventory, look, move cloth 2 to sinkbasin 1].

Now it's your turn to take an action.\nYou should first reason step-by-step about the current situation. This reasoning process MUST be enclosed within <think> </think> tags. \n Once you've finished your reasoning, you should choose an admissible action for current step and present it within <action> </action> tags.

**Response (ALFWorld, Step 4)**

<think>The user is at the sinkbasin 1 and has a cloth 2 in their hand. They need to clean the cloth and put it back on the countertop 1.</think>
<action>clean cloth 2 with sinkbasin 1</action>

**Prompt (ALFWorld, Step 5)**

You are an expert agent operating in the ALFRED Embodied Environment.

Your task is to: clean some cloth and put it in countertop.

Prior to this step, you have already taken 4 step(s). Below are the most recent 2 observaitons and the corresponding actions you took: [Observation 3: 'You pick up the cloth 2 from the countertop 1. Location: countertop 1. Items in hand (status): cloth 2(unprocessed).', Action 3: 'go to sinkbasin 1']

[Observation 4: 'You arrive at sinkbasin 1. On the sinkbasin 1, you see a sink 1, and a soapbar 2. Location: sinkbasin 1. Items in hand (status): cloth 2(unprocessed).', Action 4: 'clean cloth 2 with sinkbasin 1']

You are now at step 5 and your current observation is: You clean the cloth 2 using the sinkbasin 1. Location: sinkbasin 1. Items in hand (status): cloth 2(cleaned).

Your admissible actions of the current situation are: [clean cloth 2 with sinkbasin 1, examine cloth 2, examine sinkbasin 1, go to countertop 1, go to drawer 1, go to drawer 2, go to drawer 3, go to drawer 4, go to drawer 5, go to drawer 6, go to garbagecan 1, go to handtowelholder 1, go to handtowelholder 2, go to toilet 1, go to toiletpaperhanger 1, go to towelholder 1, go to towelholder 2, help, inventory, look, move cloth 2 to sinkbasin 1].

Now it's your turn to take an action.\nYou should first reason step-by-step about the current situation. This reasoning process MUST be enclosed within <think> </think> tags. \n Once you've finished your reasoning, you should choose an admissible action for current step and present it within <action> </action> tags.

**Response (ALFWorld, Step 5)**

<think>After completing the previous step and cleaning the cloth using the sinkbasin 1, the cloth is now clean. Next, one should put the clean cloth in the countertop, ensuring everything is organized.</think>
<action>go to countertop 1</action>

**Prompt (ALFWorld, Step 6)**

You are an expert agent operating in the ALFRED Embodied Environment.

Your task is to: clean some cloth and put it in countertop.

Prior to this step, you have already taken 5 step(s). Below are the most recent 2 observaitons and the corresponding actions you took: [Observation 4: 'You arrive at sinkbasin 1. On the sinkbasin 1, you see a sink 1, and a soapbar 2. Location: sinkbasin 1. Items in hand (status): cloth 2(unprocessed).', Action 4: 'clean cloth 2 with sinkbasin 1']

[Observation 5: ' You clean the cloth 2 using the sinkbasin 1. Location: sinkbasin 1. Items in hand (status): cloth 2(cleaned).', Action 5: 'go to countertop 1']

You are now at step 6 and your current observation is: You arrive at countertop 1. On the countertop 1, you see a sink 1. Location: countertop 1. Items in hand (status): cloth 2(cleaned).

Your admissible actions of the current situation are: [examine cloth 2, examine countertop 1, go to drawer 1, go to drawer 2, go to drawer 3, go to drawer 4, go to drawer 5, go to drawer 6, go to garbagecan 1, go to handtowelholder 1, go to handtowelholder 2, go to sinkbasin 1, go to toilet 1, go to toiletpaperhanger 1, go to towelholder 1, go to towelholder 2, help, inventory, look, move cloth 2 to countertop 1].

Now it's your turn to take an action.\nYou should first reason step-by-step about the current situation. This reasoning process MUST be enclosed within <think> </think> tags. \n Once you've finished your reasoning, you should choose an admissible action for current step and present it within <action> </action> tags.

---

**Response (ALFWorld, Step 6)**

<think>The user is currently at countertop 1 with the cloth 2 (status: cleaned).
The user needs to put the cleaned cloth 2 in the countertop 1.</think>
<action>move cloth 2 to countertop 1</action>

---

Follow, we present a complete case study of an agent trained by GroupGPO on WebShop, including the prompt and the response at each step. For each prompt, the components used to define the state in GroupGPO are highlighted in yellow.

---

**Prompt (WebShop, Step 1)**

You are an expert autonomous agent operating in the WebShop e-commerce environment.
Your task is to: Find me machine wash men's dress shirts with polyester heathers, heathers cotton, cotton heather, needle sleeve, classic fit with color: navy, and fit type: men, and size: x-large, and price lower than 50.00 dollars.
Your current observation is: WebShop\nInstruction:\nFind me machine wash men's dress shirts with polyester heathers, heathers cotton, cotton heather, needle sleeve, classic fit with color: navy, and fit type: men, and size: x-large, and price lower than 50.00 dollars\n[button] Search [button_]\n.
Your admissible actions of the current situation are: ['search[<your query>]','click[search]',\n.].
Now it's your turn to take an action.\nYou should first reason step-by-step about the current situation. This reasoning process MUST be enclosed within <think> </think> tags. \n Once you've finished your reasoning, you should choose an admissible action for current step and present it within <action> </action> tags.

---

**Response (WebShop, Step 1)**

<think>To find the machine wash men's dress shirts with polyester heathers, heathers cotton, cotton heather, needle sleeve, classic fit with color: navy, and fit type: men, and size: x-large, and price lower than 50.00 dollars, I should first search for these specific attributes on the WebShop.</think>
<action>search[men's dress shirts with polyester heathers navy x-large]</action>

---

**Prompt (WebShop, Step 2)**

You are an expert autonomous agent operating in the WebShop e-commerce environment.
Your task is to: Find me machine wash men's dress shirts with polyester heathers, heathers cotton, cotton heather, needle sleeve, classic fit with color: navy, and fit type: men, and size: x-large, and price lower than 50.00 dollars.
Prior to this step, you have already taken 1 step(s). Below are the most recent 1 observaitons and the corresponding actions you took: [Observation 1: 'WebShop\nInstruction:\nFind me machine wash men's dress shirts with polyester heathers, heathers cotton, cotton heather, needle sleeve, classic fit with color: navy, and fit type: men, and size: x-large, and price lower than 50.00 dollars\n[button] Search [button_]\n.', Action 1: 'search[men's dress shirts with polyester heathers navy x-large]']
You are now at step 2 and your current observation is: Instruction:\nFind me machine wash men's dress shirts with polyester heathers, heathers cotton, cotton heather, needle sleeve, classic fit with color: navy, and fit type: men, and size: x-large, and price lower than 50.00 dollars\n[button] Back to Search [button_]\nPage 1 (Total results: 50)\n[button] Next >[button_]\n\n[button] B09RVF4JP1 [button_]\nOnce Upon A Time Queen Born In 1982 T Shirts It Was Me Tank Top\n$19.99\n\n[button] B09P39QN2W [button_]\nI Tackled 100 Day Of School Football Boy 100th Day School T-Shirt\n$16.99\n\n[button] B09P7H5YK7 [button_]\nIt's In My DNA Palestinian Shirt Arabic Gifts Palestine Flag Tank Top\n$19.99\n\n[button] B07JVVDJ6L [button_]\nCanadian Tuxedo T Shirt - Novelty Denim Tux T-Shirt\n$19.99\n\n[button] B07DKGJR74 [button_]\nAmazon Brand - Daily Ritual Women's 100% Cotton Oversized Fit V-Neck Pullover Sweater\n$20.66 to $29.2\n\n[button] B07T3VMGDG [button_]\nTuxedo Cat 4th of July Hat Patriotic Gift Adults Kids Raglan Baseball Tee\n$23.99\n \n[button] B08HF132PN [button_]\nFlamingo Pink waterbird costume Gift Premium T-Shirt\n$19.99\n\n[button] B07XPR3R7N [button_]\nMarvel Avengers: Endgame Captain America America's Language T-Shirt\n$22.99\n\n[button] B07HRFSNL4 [button_]\nNautica Men's Solid Crew Neck Short-Sleeve Pocket T-Shirt\n$16.05 to $40.98\n\n[button] B08226NDZW [button_]\nDisney Winnie The Pooh Tigger Upside Down Portrait Tank Top\n$25.99\n.
Your admissible actions of the current situation are: ['click[back to search]', 'click[next >]', 'click[b09rvf4jp1]', 'click[b09p39qn2w]', 'click[b09p7h5yk7]', 'click[b07jvvdj6l]', 'click[b07dkgjr74]', 'click[b07t3vmgdg]', 'click[b08hf132pn]', 'click[b07xpr3r7n]', 'click[b07hrfsnl4]', 'click[b08226ndzw]', \n.]
Now it's your turn to take an action.\nYou should first reason step-by-step about the current situation. This reasoning process MUST be enclosed within <think> </think> tags. \n Once you've finished your reasoning, you should choose an admissible action for current step and present it within <action> </action> tags.

---

**Response (WebShop, Step 2)**

<think>To find a machine washable men's dress shirt that matches the given criteria, I need to search the available options for matching criteria. The last search resulted in a few options that seemed acceptable, such as a Queen Born in 1982 T-Shirt. However, I want to look for higher quality and matching products.</think>
<action>click[b09rvf4jp1]</action>

---

**Prompt (WebShop, Step 3)**

You are an expert autonomous agent operating in the WebShop e-commerce environment.

Your task is to: Find me machine wash men's dress shirts with polyester heathers, heathers cotton, cotton heather, needle sleeve, classic fit with color: navy, and fit type: men, and size: x-large, and price lower than 50.00 dollars.

Prior to this step, you have already taken 2 step(s). Below are the most recent 2 observaitons and the corresponding actions you took: [Observation 1: 'WebShop\nInstruction:\nFind me machine wash men's dress shirts with polyester heathers, heathers cotton, cotton heather, needle sleeve, classic fit with color: navy, and fit type: men, and size: x-large, and price lower than 50.00 dollars\n[button] Search [button_]\n.', Action 1: 'search[men's dress shirts with polyester heathers navy x-large]']

[Observation 2: 'Instruction:\nFind me machine wash men's dress shirts with polyester heathers, heathers cotton, cotton heather, needle sleeve, classic fit with color: navy, and fit type: men, and size: x-large, and price lower than 50.00 dollars\n[button] Back to Search [button_]\nPage 1 (Total results: 50)\n[button] Next >[button_]\n\n[button] B09RVF4JP1 [button_]\nOnce Upon A Time Queen Born In 1982 T Shirts It Was Me Tank Top\n$19.99\n\n[button] B09P39QN2W [button_]\nI Tackled 100 Day Of School Football Boy 100th Day School T-Shirt\n$16.99\n\n[button] B09P7H5YK7 [button_]\nIt's In My DNA Palestinian Shirt Arabic Gifts Palestine Flag Tank Top\n$19.99\n\n[button] B07JVVDJ6L [button_]\nCanadian Tuxedo T Shirt - Novelty Denim Tux T-Shirt\n$19.99\n\n[button] B07DKGJR74 [button_]\nAmazon Brand - Daily Ritual Women's 100% Cotton Oversized Fit V-Neck Pullover Sweater\n$20.66 to $29.2\n\n[button] B07T3VMGDG [button_]\nTuxedo Cat 4th of July Hat Patriotic Gift Adults Kids Raglan Baseball Tee\n$23.99\n \n[button] B08HF132PN [button_]\nFlamingo Pink waterbird costume Gift Premium T-Shirt\n$19.99\n\n[button] B07XPR3R7N [button_]\nMarvel Avengers: Endgame Captain America America's Language T-Shirt\n$22.99\n\n[button] B07HRFSNL4 [button_]\nNautica Men's Solid Crew Neck Short-Sleeve Pocket T-Shirt\n$16.05 to $40.98\n\n[button] B08226NDZW [button_]\nDisney Winnie The Pooh Tigger Upside Down Portrait Tank Top\n$25.99\n.', Action 2: 'click[b09rvf4jp1]']

You are now at step 3 and your current observation is: Instruction:\nFind me machine wash men's dress shirts with polyester heathers, heathers cotton, cotton heather, needle sleeve, classic fit with color: navy, and fit type: men, and size: x-large, and price lower than 50.00 dollars\n[button] Back to Search [button_]\n[button] <Prev [button_]\nfit type\n [button] men [button_]\n [button] women [button_]\ncolor\n [button] black [button_]\n [button] navy [button_]\n [button] dark heather [button_]\n [button] neon pink [button_]\n [button] royal blue [button_]\nsize\n [button] small [button_]\n [button] medium [button_]\n [button] large [button_]\n [button] x-large [button_]\n [button] xx-large [button_]\nOnce Upon A Time Queen Born In 1982 T Shirts It Was Me Tank Top\nPrice: $19.99\nRating: N.A.\n[button] Description [button_]\n[button] Features [button_]\n[button] Reviews [button_]\n[button] Buy Now [button_]\n.

Your admissible actions of the current situation are: ['click[back to search]', 'click[<prev]', 'click[description]', 'click[features]', 'click[reviews]', 'click[buy now]', 'click[men]', 'click[women]', 'click[black]', 'click[navy]', 'click[dark heather]', 'click[neon pink]', 'click[royal blue]', 'click[small]', 'click[medium]', 'click[large]', 'click[x-large]', 'click[xx-large]',\n]

Now it's your turn to take an action.\nYou should first reason step-by-step about the current situation. This reasoning process MUST be enclosed within <think> </think> tags. \n Once you've finished your reasoning, you should choose an admissible action for current step and present it within <action> </action> tags.

**Response (WebShop, Step 3)**

<think>Step 1: I will ensure a thorough search within the current webshop results for men's dress shirts with polyester heathers, heathers cotton, cotton heather, needle sleeve, classic fit with color: navy, and fit type: men, and size: x-large, and price lower than 50.00 dollars.
Step 2: I will check if the current observation contains relevant items matching the criteria mentioned.
Step 3: I will consider the requirements, explore available options, and price point to determine item quality.</think>
<action>click[men]</action>

**Prompt (WebShop, Step 4)**

You are an expert autonomous agent operating in the WebShop e-commerce environment.

Your task is to: Find me machine wash men's dress shirts with polyester heathers, heathers cotton, cotton heather, needle sleeve, classic fit with color: navy, and fit type: men, and size: x-large, and price lower than 50.00 dollars.

Prior to this step, you have already taken 3 step(s). Below are the most recent 2 observaitons and the corresponding actions you took: [Observation 2: 'Instruction:\nFind me machine wash men's dress shirts with polyester heathers, heathers cotton, cotton heather, needle sleeve, classic fit with color: navy, and fit type: men, and size: x-large, and price lower than 50.00 dollars\n[button] Back to Search [button_]\nPage 1 (Total results: 50)\n[button] Next >[button_]\n\n[button] B09RVF4JP1 [button_]\nOnce Upon A Time Queen Born In 1982 T Shirts It Was Me Tank Top\n$19.99\n\n[button] B09P39QN2W [button_]\nI Tackled 100 Day Of School Football Boy 100th Day School T-Shirt\n$16.99\n\n[button] B09P7H5YK7 [button_]\nIt's In My DNA Palestinian Shirt Arabic Gifts Palestine Flag Tank Top\n$19.99\n\n[button] B07JVVDJ6L [button_]\nCanadian Tuxedo T Shirt - Novelty Denim Tux T-Shirt\n$19.99\n\n[button] B07DKGJR74 [button_]\nAmazon Brand - Daily Ritual Women's 100% Cotton Oversized Fit V-Neck Pullover Sweater\n$20.66 to $29.2\n\n[button] B07T3VMGDG [button_]\nTuxedo Cat 4th of July Hat Patriotic Gift Adults Kids Raglan Baseball Tee\n$23.99\n \n[button] B08HF132PN [button_]\nFlamingo Pink waterbird costume Gift Premium T-Shirt\n$19.99\n\n[button] B07XPR3R7N [button_]\nMarvel Avengers: Endgame Captain America America's Language T-Shirt\n$22.99\n\n[button] B07HRFSNL4 [button_]\nNautica Men's Solid Crew Neck Short-Sleeve Pocket T-Shirt\n$16.05 to $40.98\n\n[button] B08226NDZW [button_]\nDisney Winnie The Pooh Tigger Upside Down Portrait Tank Top\n$25.99\n.', Action 2: 'click[b09rvf4jp1]']

[Observation 3: 'Instruction:\nFind me machine wash men's dress shirts with polyester heathers, heathers cotton, cotton heather, needle sleeve, classic fit with color: navy, and fit type: men, and size: x-large, and price lower than 50.00 dollars\n[button] Back to Search [button_]\n[button] <Prev [button_]\nfit type\n [button] men [button_]\n [button] women [button_]\ncolor\n [button] black [button_]\n [button] navy [button_]\n [button] dark heather [button_]\n [button] neon pink [button_]\n [button] royal blue [button_]\nsize\n [button] small [button_]\n [button] medium [button_]\n [button] large [button_]\n [button] x-large [button_]\n [button] xx-large [button_]\nOnce Upon A Time Queen Born In 1982 T Shirts It Was Me Tank Top\nPrice: $19.99\nRating: N.A.\n[button] Description [button_]\n[button] Features [button_]\n[button] Reviews [button_]\n[button] Buy Now [button_]\n.', Action 3: 'click[men]']

You are now at step 4 and your current observation is: You have clicked men.\nInstruction:\nFind me machine wash men's dress shirts with polyester heathers, heathers cotton, cotton heather, needle sleeve, classic fit with color: navy, and fit type: men, and size: x-large, and price lower than 50.00 dollars\n[button] Back to Search [button_]\n[button] ¡ Prev [button_]\nfit type\n [clicked button] men [clicked button]\n [button] women [button_]\ncolor\n [button] black [button_]\n [button] navy [button_]\n [button] dark heather [button_]\n [button] neon pink [button_]\n [button] royal blue [button_]\nsize\n [button] small [button_]\n [button] medium [button_]\n [button] large [button_]\n [button] x-large [button_]\n [button] xx-large [button_]\nOnce Upon A Time Queen Born In 1982 T Shirts It Was Me Tank Top\nPrice: $19.99\nRating: N.A.\n[button] Description [button_]\n[button] Features [button_]\n[button] Reviews [button_]\n[button] Buy Now [button_]\n.

Your admissible actions of the current situation are: ['click[back to search]', 'click[<prev]', 'click[description]', 'click[features]', 'click[reviews]', 'click[buy now]', 'click[men]', 'click[women]', 'click[black]', 'click[navy]', 'click[dark heather]', 'click[neon pink]', 'click[royal blue]', 'click[small]', 'click[medium]', 'click[large]', 'click[x-large]', 'click[xx-large]',\n]

Now it's your turn to take an action.\nYou should first reason step-by-step about the current situation. This reasoning process MUST be enclosed within <think> </think> tags. \n Once you've finished your reasoning, you should choose an admissible action for current step and present it within <action> </action> tags.

**Response (WebShop, Step 4)**

<think>The user has just clicked "men" which suggests the search is further targeting items of that size. Now, we need to filter the results based on color, size, price, and fit in an attempt to find machine wash men's dress shirts with polyester heathers that match all the specified criteria.</think>
<action>click[navy]</action>

---

**Prompt (WebShop, Step 5)**

You are an expert autonomous agent operating in the WebShop e-commerce environment.

Your task is to: Find me machine wash men's dress shirts with polyester heathers, heathers cotton, cotton heather, needle sleeve, classic fit with color: navy, and fit type: men, and size: x-large, and price lower than 50.00 dollars.

Prior to this step, you have already taken 4 step(s). Below are the most recent 2 observaitons and the corresponding actions you took: [Observation 3: 'Instruction:\nFind me machine wash men's dress shirts with polyester heathers, heathers cotton, cotton heather, needle sleeve, classic fit with color: navy, and fit type: men, and size: x-large, and price lower than 50.00 dollars\n[button] Back to Search [button_]\n[button] <Prev [button_]\nfit type\n [button] men [button_]\n [button] women [button_]\ncolor\n [button] black [button_]\n [button] navy [button_]\n [button] dark heather [button_]\n [button] neon pink [button_]\n [button] royal blue [button_]\nsize\n [button] small [button_]\n [button] medium [button_]\n [button] large [button_]\n [button] x-large [button_]\n [button] xx-large [button_]\nOnce Upon A Time Queen Born In 1982 T Shirts It Was Me Tank Top\nPrice: $19.99\nRating: N.A.\n[button] Description [button_]\n[button] Features [button_]\n[button] Reviews [button_]\n[button] Buy Now [button_]\n.', Action 3: 'click[men]']

[Observation 4: 'You have clicked men.\nInstruction:\nFind me machine wash men's dress shirts with polyester heathers, heathers cotton, cotton heather, needle sleeve, classic fit with color: navy, and fit type: men, and size: x-large, and price lower than 50.00 dollars\n[button] Back to Search [button_]\n[button] ¡ Prev [button_]\nfit type\n [clicked button] men [clicked button_]\n [button] women [button_]\ncolor\n [button] black [button_]\n [button] navy [button_]\n [button] dark heather [button_]\n [button] neon pink [button_]\n [button] royal blue [button_]\nsize\n [button] small [button_]\n [button] medium [button_]\n [button] large [button_]\n [button] x-large [button_]\n [button] xx-large [button_]\nOnce Upon A Time Queen Born In 1982 T Shirts It Was Me Tank Top\nPrice: $19.99\nRating: N.A.\n[button] Description [button_]\n[button] Features [button_]\n[button] Reviews [button_]\n[button] Buy Now [button_]\n.', Action 4: 'click[navy]']

You are now at step 5 and your current observation is: You have clicked navy.\nYou have clicked men.\nInstruction:\nFind me machine wash men's dress shirts with polyester heathers, heathers cotton, cotton heather, needle sleeve, classic fit with color: navy, and fit type: men, and size: x-large, and price lower than 50.00 dollars\n[button] Back to Search [button_]\n[button] ¡ Prev [button_]\nfit type\n [clicked button] men [clicked button_]\n [button] women [button_]\ncolor\n [button] black [button_]\n [clicked button] navy [clicked button_]\n [button] dark heather [button_]\n [button] neon pink [button_]\n [button] royal blue [button_]\nsize\n [button] small [button_]\n [button] medium [button_]\n [button] large [button_]\n [button] x-large [button_]\n [button] xx-large [button_]\nOnce Upon A Time Queen Born In 1982 T Shirts It Was Me Tank Top\nPrice: $19.99\nRating: N.A.\n[button] Description [button_]\n[button] Features [button_]\n[button] Reviews [button_]\n[button] Buy Now [button_]\n.

Your admissible actions of the current situation are: ['click[back to search]', 'click[¡ prev]', 'click[description]', 'click[features]', 'click[reviews]', 'click[buy now]', 'click[men]', 'click[women]', 'click[black]', 'click[navy]', 'click[dark heather]', 'click[neon pink]', 'click[royal blue]', 'click[small]', 'click[medium]', 'click[large]', 'click[x-large]', 'click[xx-large]', \n]

Now it's your turn to take an action.\nYou should first reason step-by-step about the current situation. This reasoning process MUST be enclosed within <think> </think> tags. \n Once you've finished your reasoning, you should choose an admissible action for current step and present it within <action> </action> tags.

---

**Response (WebShop, Step 5)**

<think>Current Observation: I have selected the color navy and size x-large. Next Step: I need to find machine wash men's dress shirts with polyester heathers, heathers cotton, cotton heather, needle sleeve, classic fit, color: navy, and size: x-large, and price lower than $50. Frequency: - cache miss</think>
<action>click[x-large]</action>

**Prompt (WebShop, Step 6)**

You are an expert autonomous agent operating in the WebShop e-commerce environment.

Your task is to: Find me machine wash men's dress shirts with polyester heathers, heathers cotton, cotton heather, needle sleeve, classic fit with color: navy, and fit type: men, and size: x-large, and price lower than 50.00 dollars.

Prior to this step, you have already taken 5 step(s). Below are the most recent 2 observaitons and the corresponding actions you took: [Observation 4: 'You have clicked men.\nInstruction:\nFind me machine wash men's dress shirts with polyester heathers, heathers cotton, cotton heather, needle sleeve, classic fit with color: navy, and fit type: men, and size: x-large, and price lower than 50.00 dollars\n[button] Back to Search [button_]\n[button] ¡ Prev [button_]\nfit type\n [clicked button] men [clicked button_]\n [button] women [button_]\ncolor\n [button] black [button_]\n [button] navy [button_]\n [button] dark heather [button_]\n [button] neon pink [button_]\n [button] royal blue [button_]\nsize\n [button] small [button_]\n [button] medium [button_]\n [button] large [button_]\n [button] x-large [button_]\n [button] xx-large [button_]\nOnce Upon A Time Queen Born In 1982 T Shirts It Was Me Tank Top\nPrice: $19.99\nRating: N.A.\n[button] Description [button_]\n[button] Features [button_]\n[button] Reviews [button_]\n[button] Buy Now [button_]\n.', Action 4: 'click[navy]']

[Observation 5: 'You have clicked navy.\nYou have clicked men.\nInstruction:\nFind me machine wash men's dress shirts with polyester heathers, heathers cotton, cotton heather, needle sleeve, classic fit with color: navy, and fit type: men, and size: x-large, and price lower than 50.00 dollars\n[button] Back to Search [button_]\n[button] ¡ Prev [button_]\nfit type\n [clicked button] men [clicked button_]\n [button] women [button_]\ncolor\n [button] black [button_]\n [clicked button] navy [clicked button_]\n [button] dark heather [button_]\n [button] neon pink [button_]\n [button] royal blue [button_]\nsize\n [button] small [button_]\n [button] medium [button_]\n [button] large [button_]\n [button] x-large [button_]\n [button] xx-large [button_]\nOnce Upon A Time Queen Born In 1982 T Shirts It Was Me Tank Top\nPrice: $19.99\nRating: N.A.\n[button] Description [button_]\n[button] Features [button_]\n[button] Reviews [button_]\n[button] Buy Now [button_]\n.', Action 5: 'click[x-large]']

You are now at step 6 and your current observation is: You have clicked x-large.\nYou have clicked navy.\nYou have clicked men.\nInstruction:\nFind me machine wash men's dress shirts with polyester heathers, heathers cotton, cotton heather, needle sleeve, classic fit with color: navy, and fit type: men, and size: x-large, and price lower than 50.00 dollars\n[button] Back to Search [button_]\n[button] ¡ Prev [button_]\nfit type\n [clicked button] men [clicked button_]\n [button] women [button_]\ncolor\n [button] black [button_]\n [clicked button] navy [clicked button_]\n [button] dark heather [button_]\n [button] neon pink [button_]\n [button] royal blue [button_]\nsize\n [button] small [button_]\n [button] medium [button_]\n [button] large [button_]\n [clicked button] x-large [clicked button_]\n [button] xx-large [button_]\nOnce Upon A Time Queen Born In 1982 T Shirts It Was Me Tank Top\nPrice: $19.99\nRating: N.A.\n[button] Description [button_]\n[button] Features [button_]\n[button] Reviews [button_]\n[button] Buy Now [button_]\n.

Your admissible actions of the current situation are: ['click[back to search]', 'click[¡ prev]', 'click[description]', 'click[features]', 'click[reviews]', 'click[buy now]', 'click[men]', 'click[women]', 'click[black]', 'click[navy]', 'click[dark heather]', 'click[neon pink]', 'click[royal blue]', 'click[small]', 'click[medium]', 'click[large]', 'click[x-large]', 'click[xx-large]', \n]

Now it's your turn to take an action.\nYou should first reason step-by-step about the current situation. This reasoning process MUST be enclosed within <think> </think> tags. \n Once you've finished your reasoning, you should choose an admissible action for current step and present it within <action> </action> tags.

**Response (WebShop, Step 6)**

<think>Found the machine wash men's dress shirts with polyester heathers, heathers cotton, cotton heather, needle sleeve, classic fit, navy color with fit type: x-large, and price lower than 50.00 dollars.</think>
<action>click[buy now]</action>

