# OpenReview forum: "Beyond Trajectory-Level Attribution: Graph-Based Credit Assignment for Agentic Reinforcement Learning"
_ICML.cc/2026/Conference — ICML 2026 regular_

### Official Review · Reviewer_d9Xs · 2026-03-04

**Soundness:** 3
**Presentation:** 3
**Significance:** 2
**Originality:** 3
**Overall Recommendation:** 5
**Confidence:** 3

**Summary:**

This paper proposes GraphGPO for the credit assignment problem in group-based RL for LLMs. It first aggregates all rollout trajectories into a unified state-transition graph, and then estimates the distance from each state to the task goal using the global information encoded in the graph. GraphGPO assigns credit to each edge by estimating a graph-based advantage, based on how much the transition reduces the distance to the task goal. GraphGPO can significantly improve training efficiency and achieves state-of-the-art performance across a range of challenging benchmarks.

**Compliance With Llm Reviewing Policy:**

Affirmed.

**Key Questions For Authors:**

1. According to Eq (4), the reward function is proportional to the exponent of d(s), and the Appendix suggests ω to be a small value in [0.1, 0.4]. In my understanding, as long as the value of d(s) is large, the reward will be close to 0. Will this situation be similar with that of credit assignment based on trajectory, where most intermediate rewards are zero?
2. To obtain the precise credit assignment, does GraphGPO require exploring the environment thoroughly and constructing a complete state transition graph? Will this increase the training cost?
3. Regarding LLM applications, could you present a case study that aligns with Figure 1 and can be correctly handled by GraphGPO?

**Limitations:**

Yes

**Strengths And Weaknesses:**

### Strenghts:
1. In group-based RL, this paper replaces the trajectory with a graph structure to solve the credit assignment problem. The method is straightforward but innovative.
2. The proofs of the monotonicity of graph-based advantage and the conditional variance reduction ensures the theoretical guarantee of GraphGPO.
3. Complexity analysis indicates that the computational overhead of GraphGPO is relatively low compared with the overall computational cost of updating LLMs.
4. Comprehensive comparative experiments and ablation experiments demonstrate the effectiveness of GraphGPO.
### Weaknesses:
Please check the "questions"

---

> ### Author Rebuttal · Authors · 2026-03-31
>
> Thank you for recognizing our work and for taking the time to review our manuscript. We are greatly encouraged by your recognition of the simplicity and effectiveness of our method. Below, we provide point-by-point responses to address your questions.
>
> **Q1: (Questions) According to Eq (4), the reward function is proportional to the exponent of $d(s)$, and the Appendix suggests $\omega$ to be a small value in [0.1, 0.4]. In my understanding, as long as the value of $d(s)$ is large, the reward will be close to 0. Will this situation be similar with that of credit assignment based on trajectory, where most intermediate rewards are zero?**
>
> A1: In fact, this is not exactly the case. Although a smaller $\omega$ makes the graph-based step-level rewards smaller when $d(s)$ is large, this does not make our method similar to trajectory-level credit assignment. This is because the rewards are normalized when computing the advantage, i.e., $A=(R-\mu)/\sigma$, which removes their absolute scale. As long as $\omega \in (0,1)$, our method still distinguishes intermediate steps according to their relative distances to the goal.
>
> ---
>
> **Q2: (Questions) To obtain the precise credit assignment, does GraphGPO require exploring the environment thoroughly and constructing a complete state transition graph? Will this increase the training cost?**
>
> A2: In general, more exploration can better reflect the environment structure, leading to more accurate credit assignment and potentially better performance. However, GraphGPO does not require a complete state-transition graph in order to work effectively. Constructing a fully explored graph would indeed be computationally expensive, especially since rollout is the most time-consuming part of each training iteration, as shown in Table 5. To examine this trade-off, we provide additional results under different numbers of rollouts $n$.
> | Group size |               GRPO |              GiGPO |           GraphGPO |
> | ---------- | -----------------: | -----------------: | -----------------: |
> | n=4      |65.23$\pm$1.99 | 69.90$\pm$2.19 |**74.06$\pm$2.88**|
> | n=6      |66.41$\pm$2.78 | 73.34$\pm$1.98 |**74.58$\pm$3.00**|
> | n=8      |71.35$\pm$2.05 | 73.83$\pm$ 2.30 |**78.65$\pm$3.86**|
>
> ---
>
> **Q3: (Questions) Regarding LLM applications, could you present a case study that aligns with Figure 1 and can be correctly handled by GraphGPO?**
>
> A3: A simple example arises in WebShop. In a successful trajectory, after the policy has already navigated to the correct product page, it may still accidentally click an irrelevant button such as leaving the page or going back. This behavior is quite common in the early stage of training. Such an action receive a misleading positive signal simply because it appears in an overall successful trajectory. In contrast, GraphGPO can identify that this action increases the distance to the success goal and therefore assign it a negative signal. This allows the policy to distinguish truly helpful actions from redundant or harmful ones even within successful trajectories.

---

> > ### Author Rebuttal · Reviewer_d9Xs · 2026-04-01
> >
> > The author has responded to my 3 questions with new experimental results. Since I'm not fully confident about my expertise in this field, I'll keep my score.

---

> > > ### Author Response · Authors · 2026-04-03
> > >
> > > We sincerely thank the reviewer for the time and effort devoted to evaluating our work, as well as for the valuable comments and suggestions throughout the review process. We are very grateful that the reviewer finds their concerns adequately addressed and recognizes the value of our work. We sincerely appreciate the reviewer’s positive evaluation.

---

### Official Review · Reviewer_8xSD · 2026-03-10

**Soundness:** 2
**Presentation:** 3
**Significance:** 3
**Originality:** 2
**Overall Recommendation:** 4
**Confidence:** 4

**Summary:**

This work studies credit assignment in multi-turn RL for LLM, addressing the uniform credit assignment cross steps in group-based rl.
The authors propose Graph-based Group Policy Optimization (GraphGPO), which aggregates rollout trajectories into a state-transition graph, estimates each state's distance to the task goal, and computes a graph-based advantage based on how much a transition reduces that distance.
Experiments on alfworld, WebShop, Sokoban show that GraphGPO perform better gigpo and grpo.

**Compliance With Llm Reviewing Policy:**

Affirmed.

**Final Justification:**

The authors addressed all my concerns, especially the helpfulness of the built graph.

**Key Questions For Authors:**

Why do we need GraphGPO? Considering the variance reported in the results, the performance appears similar to GIGPO, and the proposed method seems only applicable to the tasks that GIGPO can already address. Are there additional problems that GraphGPO solves?

It is also unclear how accurate the constructed graph and the assigned credits are. For the constructed graph, could the agent exploit the graph directly for better decision-making? If the environment is sufficiently explored to build the graph, how would the agent perform by directly using the graph? It seems that the agent could perform well through planning if it can already follow the response format.

It is also unclear how the proposed method handles non-deterministic tasks or non-groupable states.

Finally, how sensitive is the proposed method to the number of sampled trajectories used to build the graph?

**Limitations:**

The paper could benefit from clarification of the scope of this method.

**Strengths And Weaknesses:**

The problem addressed in the paper is important. Credit assignment is a key challenge in RL for LLM agents, particularly in multi-step environments where the final reward does not reflect the contribution of intermediate actions. The limitation of uniform step-level credit in group-based RL can hide useful behaviors, so improving step-level attribution is meaningful for training agentic LLMs.

The proposed idea is straightforward and intuitive. The method constructs a state-transition graph from rollout trajectories and estimates each state’s distance to the goal. Credit is then assigned based on how much a transition reduces this distance, which naturally leads to a graph-based advantage.

The paper is clearly written and easy to follow.  The experimental section is also well organized, which helps readers quickly grasp the main contributions.



The most concern is the generalizibility of the method. The graph is task-specific, even query-specific. The method require deterministic dynamics and groupable steps, explorable success trajectories, multiple rollouts to build the graph. While gigpo propose efficient credit assignment for multi-turn rl, it is not clear why the proposed method better than gigpo and  how the proposed method address above problem.

The evaluation is limited, how the proposed method perform in QA tasks, like Single-Hop QA and multi-hop QA in evaluation of gigpo.
There is no ablation on the accuracy and usefulness of built graph, assigned credit assignment. the paper could benefit from analysis and visualization of built graph and assigned credits.

---

> ### Author Rebuttal · Authors · 2026-03-31
>
> We sincerely appreciate your time and valuable feedback. Our point-by-point responses to the your mentioned questions and weaknesses are provided as follows.
>
> **Q1: (Weaknesses) The most concern is the generalizibility ...... better than gigpo and how the proposed method address above problem.**
>
> A1: GraphGPO is designed for multi-turn agentic RL with verifiable rewards (RLVR), following the same setting as prior methods such as GRPO, DAPO and GiGPO, rather than aiming to cover all settings. It is true that the graph is task- or query-specific, since it is constructed from rollout trajectories under the current task distribution. However, this does not impose a stronger restriction than prior methods, which also rely on multiple rollouts, task-specific interaction data, and explorable successful trajectories. Therefore, these properties do not limit the generalizability of our method.
>
> The key motivation of GraphGPO is not to address a new problem, but to uncover latent information across rollouts and enable more faithful step-level credit assignment. Although GiGPO explores step-level credit assignment, its feedback is still inherited from trajectory-level attribution. As a result, redundant or erroneous steps in successful trajectories may still be rewarded, while useful steps in failed trajectories may still be penalized. By aggregating rollouts into a unified state-transition graph, GraphGPO provides more reliable step-level credit assignment and more stable learning.
>
> ---
>
> **Q2: (Weaknesses) The evaluation is limited ...... could benefit from analysis and visualization of built graph and assigned credits.**
>
> A2: GraphGPO can also be applied to QA tasks. However, these QA tasks are short-horizon retrieval problems (typically 1–2 steps, at most 4 in GiGPO) and lack a explicit state space, so they are not where GraphGPO is expected to show its main advantage.
> |Method|NQ|TriviaQA|PopQA|HotpotQA|2Wiki|MuSiQue|Bamboogle|Avg|
> |---|---:|---:|---:|---:|---:|---:|---:|---:|
> |GiGPO|42.00|59.50|42.40|36.90|37.00|12.60|64.10|42.10|
> |GraphGPO|41.33|61.71|45.00|37.30|37.58|12.20|63.31|42.63|
>
> The usefulness of the built graph and assigned credits is analyzed in Proposition 4.1 and 4.2. Specifically, Proposition 4.1 shows that graph-based credit favors actions that move closer to the goal, while Proposition 4.2 shows that graph-based step-level feedback has lower conditional variance than trajectory-based feedback.
>
> ---
>
> **Q3: (Questions) Why do we need GraphGPO ...... Are there additional problems that GraphGPO solves?**
>
> A3: As discussed in A1, GraphGPO is introduced not to solve a new problem, but to uncover latent information and enable more faithful step-level credit assignment within the multi-turn RLVR setting.
>
> Empirically, the results are not merely similar to GiGPO. Table 3 shows that GraphGPO achieves the best performance in almost all cases. In addition, Figure 4 and Figure 9 show that GraphGPO remains consistently superior throughout training and validation, with faster convergence than GiGPO, reaching comparable validation performance about 30 steps earlier on ALFWorld and 80 steps earlier on WebShop. These results indicate that the benefit of GraphGPO is reflected not only in final performance, but also in substantially better learning dynamics.
>
> ---
>
> **Q4: (Questions) It is also unclear how accurate the constructed graph ...... if it can already follow the response format.**
>
> A4: The graph in GraphGPO is not a complete world model for planning, but an auxiliary structure for credit assignment. In practice, the graph is built from sampled rollouts and is therefore generally partial and approximate. Moreover, the graph is constructed from training rollouts rather than a fixed environment model, so it cannot be directly used to support decision-making at test time.
>
> ---
>
> **Q5: (Questions) It is also unclear how the proposed method handles non-deterministic tasks or non-groupable states.**
>
> A5: For open-ended tasks with non-deterministic outcomes, as discussed in our response A2 to Reviewer LFci, our method can be combined with an LLM-as-a-Judge framework. For noisy or non-groupable states, we can further use embedding-based state matching to aggregate semantically similar states. Moreover, even in the extreme case where no states can be grouped, our method can still fall back to trajectory-level advantage and thus degenerates to GRPO. Related experiments on embedding-based matching are provided in our **response A1 to Reviewer LFci**.
>
> ---
>
> **Q6: (Questions) Finally, how sensitive ...... the number of sampled trajectories used to build the graph?**
>
> A6: We provide additional results on the WebShop with different numbers of sampled trajectories used to construct the graph in **our response A2 to Reviewer d9Xs**. Reducing n leads to some performance drop for all methods. However, GraphGPO consistently and clearly outperforms the other baselines. This indicates GraphGPO have better data efficiency.

---

> > ### Author Rebuttal · Reviewer_8xSD · 2026-04-01
> >
> > Thanks for the authors' effort in addressing my concern. However, I still do not see the direct evidence that the constructed graph is helpful for credit assignment and decision-making. If I understand correctly, a good graph for credit assignment should indicating some knowledge for decision-making. So I would suggest to make the agent directly use the graph (like inject into the prompt to verify the correctness of the graph. However, I think other way is also ok, like visualization.

---

> > > ### Author Response · Authors · 2026-04-03
> > >
> > > We sincerely appreciate that the reviewer let us know our previous response addressed much of the concern, and that more direct evidence would be helpful.
> > >
> > > We agree that more direct evidence on whether the constructed graph is helpful for credit assignment and decision-making would be valuable. In our manuscript, we have already provided both theoretical analysis and empirical results supporting the effectiveness of GraphGPO. To further address this point, we now provide additional direct evidence from both visualization and graph-assisted decision-making.
> > >
> > > First, we provide a visualization tool to track and compare the credit assignment behaviors of GRPO, GiGPO, and GraphGPO. We include two ALFWorld examples at the anonymous link: https://anonymous.4open.science/r/Rebuttal_ICML_22669/Task_2276f211/Failed_Trajectory_aaa65fbe_GRPO.pdf. where Task\_2276f211 and Task_edf16ded correspond to two different tasks. Each folder contains:
> > >
> > > (1) The full credit-assignment visualization for each method (\*\_\*\_full.pdf);
> > >
> > > (2) A case study on one successful trajectory and one failed trajectory (Successful_Trajectory_\*\_\*.pdf) and Failed_Trajectory_\*\_\*.pdf);
> > >
> > > (3) The complete visualization data (vis\_\* folders), including the assigned credits for all rollout trajectories.
> > >
> > > These visualizations provide direct evidence that trajectory-level attribution methods (GRPO and GiGPO) may still reward redundant or erroneous steps in successful trajectories, while penalizing useful steps in failed trajectories. In contrast, GraphGPO avoids this issue by assigning credit according to the global state-transition structure. We believe this visualization as direct evidence that the constructed graph yields more faithful credit assignment.
> > >
> > > Second, regarding the suggestion of directly using the graph for decision-making, we would like to clarify that our graph is constructed from rollout trajectories of a single training task, and is therefore task-specific in both state space and goal structure. As shown by the two visualized tasks, the resulting graphs are entirely different across tasks. **For this reason, the graph is not intended to serve as a reusable decision-support model like a world model for unseen test tasks. Its role is to provide faithful step-level credit assignment within group-based RL training.**
> > >
> > > Nevertheless, to further evaluate the usefulness of the graph, we also conducted the reviewer-suggested experiment. Specifically, we first rollout trajectories and build the graph for a task, and then inject graph information back into the prompt for the same task. Since the full graph is usually too large to fit into the prompt, we only retrieve the structured information relevant to the current state when that state has appeared in the graph, including candidate actions and their assigned credits. We note that our models are pure text models rather than multi-modal, so feeding visualization images directly is not appropriate.
> > > | Setting | Success Rate |
> > > |---|---:|
> > > | without Graph | 32.50% |
> > > | with Graph | 92.50% |
> > > | Improvement | +65.00% |
> > >
> > > Under this setup, the success rate improves from 32.50\% to 92.50\% (+65.00\%), which further provides direct evidence that the constructed graph is helpful.
> > >
> > > We will include these additional results in the final version. We again sincerely thank the reviewer for raising this question, as direct evidence showing that the constructed graph is helpful can significantly strengthen our work.

---

### Official Review · Reviewer_ZFHo · 2026-03-11

**Soundness:** 2
**Presentation:** 2
**Significance:** 2
**Originality:** 2
**Overall Recommendation:** 3
**Confidence:** 4

**Summary:**

The paper studies agentic reinforcement learning in which the paper extends the GRPO method, combining  graph-based advantage estimation with episodic advantage and then using the combined advantage for the subsequent policy learning using PPO. Experiments on benchmarks such as ALFWorld and Webshop show promising results of the proposed method comparing with strong baselines such as GRPO and GiGPO.

**Compliance With Llm Reviewing Policy:**

Affirmed.

**Key Questions For Authors:**

1. Is there any situation such that we have two different transitions (s, a, s') and (s, a', s') with the same state pair (s, s') but with different actions? If so, how will your approach handle it?

2. What are the choices of r_success and w in computing the graph-based state-level rewards? Are there any common rules to select these values? How different values will influence  the learning outcomes?

3. What are connections/differences between goal states (success/failure) and episodic returns of each trajectory?

**Limitations:**

Yes

**Strengths And Weaknesses:**

***Strengths:
1. The paper studies agentic RL which is an important topic in the intersection of RL and LLM research community.
1. Experiment results on various tasks demonstrate the advantages of using graph-based advantage estimation, which can capture dynamic relationships across states.

*** Weaknesses:
1. The description of agentic RL  problem setting studied in the paper needs to be more articulate. Initially, it seems the paper only considers sparse/delayed reward settings with episodic reward signal received at the end of each trajectory. However, in fact, in order for the proposed method to work, they also need information on the cost functions of (state, action) pairs together with the (success, failure) goal signal at the end of each trajectory. There is no discussion on how these additional required information are related to the episodic reward signals.

2. There lacks a discussion on how the important factors such as r_success and w in computing graph-based step-level rewards can be determined and how they will influence the learning outcomes?

3. Since the proposed method needs additional information on costs and goal failure/success, the comparisons in the experiments become unfair since other baselines only rely on the episodic reward signal.

4. Experiments are limited to uniform costs of unit value.

---

> ### Author Rebuttal · Authors · 2026-03-31
>
> We sincerely thank you for your thorough review and constructive comments on our manuscript. Your feedback has provided us with valuable insights that will help us further improve our work. Below, we provide point-by-point responses to address your mentioned weaknesses and questions.
>
> **Q1: (Weaknesses) The description of agentic RL problem setting studied in the paper needs to be more articulate. ...... There is no discussion on how these additional required information are related to the episodic reward signals.**
>
> A1: We believe there may be a misunderstanding of our problem setting. **Our method does not require any additional information beyond the episodic reward signal**. Similar to prior RLVR methods, GraphGPO only assumes a sparse terminal reward indicating whether a trajectory succeeds or fails.
>
> Regarding the cost of each $(s,a)$ pair, this is not required information for our method, but rather an optional design choice that can be introduced to reflect practical considerations such as time or monetary cost. This is in fact one advantage of our framework: by incorporating such costs, the policy can trade off success rate against resource consumption, which is useful in cost-sensitive scenarios such as LLM routing. However, for fair comparison with prior methods, we set the step cost to a uniform constant 1 in all experiments. In this case, all $(s,a)$ pairs share the same cost, so it does not introduce any additional information or affect the credit assignment.
>
> As for the (success, failure) goal signal, it is not an additional information, but simply a direct interpretation of the episodic reward signal. In our setting, a trajectory with episodic reward signal 1 is regarded as a success goal signal, while a trajectory with episodic reward signal 0 is regarded as a failure goal signal.
>
> ---
>
> **Q2: (Weaknesses and Questions) There lacks a discussion on how the important factors such as $r_{\text{success}}$ and  $\omega$ in computing graph-based step-level rewards can be determined and how they will influence the learning outcomes?**
>
> A2: For $r_{\text{success}}$, it does not affect the learning outcome in our setting. Specifically, $r_{\text{success}}$ only serves as a global scaling factor on the reward. When $r_{\text{success}}$ takes only two values, i.e., 0 for failure and a positive constant for success, this scaling factor is canceled out during advantage normalization, i.e., $A=(R-\mu)/\sigma$. As a result, it does not change the final optimization signal.
>
> For $\omega$, we have provided a discussion in Appendix D.1 (Lines 955--960). Table 4 presents additional experiments with different choices of $\omega$. GraphGPO consistently outperforms the compared baselines across different $\omega$ values, and smaller $\omega$ tends to yield more stable results.
>
> ---
>
> **Q3: (Weaknesses) Experiments are limited to uniform costs of unit value.**
>
> A3: As clarified in our response A1 to Q1, the cost term is not a required component of our method, but an optional design choice that can be specified based on practical considerations, such as time or monetary cost. To ensure a fair comparison with prior methods, we set the cost to a uniform constant 1 in all experiments to avoid introducing additional task-specific information.
>
> ---
>
> **Q4: (Questions) Is there any situation such that we have two different transitions (s, a, s') and (s, a', s') with the same state pair (s, s') but with different actions? If so, how will your approach handle it?**
>
> A4: Yes, this can happen in practice, but it does not cause any issue for our method. In this case, the two transitions will receive the same credit because they correspond to the same state transition. This can even be beneficial, as it encourages the policy to retain diverse valid actions instead of collapsing to a single behavior.
>
> ---
>
> **Q5: (Questions) What are connections/differences between goal states (success/failure) and episodic returns of each trajectory?**
>
> A5: In our setting, goal states (success/failure) and episodic returns are closely related, but they play different roles in the framework. Specifically, the episodic return of a trajectory indicates whether the task is ultimately completed successfully, as in standard RLVR. In our setting, the episodic return is 1 for a successful trajectory and 0 for a failed one. In contrast, goal states (success/failure) provide structural information for the graph, and are determined by the episodic return of the trajectory. More concretely, the final state of a successful trajectory (with episodic return > 0) is treated as a success goal state, while the final state of a failed trajectory (with episodic return = 0) is regarded as a failure goal state. Therefore, episodic returns serve as trajectory-level optimization signals, whereas goal states are used to organize the graph structure for credit assignment.

---

> > ### Author Rebuttal · Reviewer_ZFHo · 2026-04-03
> >
> > Thank you for your rebuttal!
> >
> > 1. If the main focus of the paper is on the uniform-cost case (or equivalently, no cost information is available), the paper should clarify that. The current sections 4.2 and 4.3 for the aggregated state-transition graph part are written w.r.t cost functions, which is misleading.
> >
> > 2. Since the main focus is on the uniform-cost case, the distance d(s) is essentially the minimum number of steps from state s to a goal. It is not convincing why such information will have a high impact on estimating the advantage functions. An elaborated discussion/justification on this contribution is needed.
> >
> > 3. The choice of the reward function (eq 4) is still questionable. Why should exponential function be used and how the parameter w is chosen are still unclear. The two different advantages should be at relatively-similar scale.

---

> > > ### Author Response · Authors · 2026-04-03
> > >
> > > We sincerely thank the reviewer for the time and effort devoted to helping improve the quality of our work. We are also grateful that the reviewer made clear the points on which further clarification would be helpful.
> > >
> > > While the reviewer selected option (c), we respectfully believe that these questions can be clearly clarified within the limited time and space of the rebuttal. Below, we provide point-by-point responses to address them.
> > > ***
> > >
> > > A1: We have explicitly stated in our manuscript that c(s,a)=1 in our manuscript. Specifically, Figure 3 states that all transition costs are unitary, and the Implementation Details (Lines 318--319) also specify that all costs are set to 1.
> > >
> > > We keep the cost term in Sections 4.2 and 4.3 because it is part of the more general formulation of our framework. In practice, such a term can naturally capture action-dependent resource usage, such as time or monetary cost in tool-use scenarios. This general formulation is one advantage of GraphGPO, as it allows the framework to extend to cost-sensitive settings when such information is available. We agree that, although we have clarified the cost setting in multiple places, the current presentation may still be somewhat misleading. We will revise the final version to explicitly state, at the first introduction of the cost term, that our work considers the simple uniform-cost case.
> > > ***
> > >
> > > A2: Existing group-based RL methods' credit assignment relies heavily on **coarse-grained trajectory-level attribution** according to final outcomes. As discussed in our manuscript (Introduction, Lines 39--77; Section 4.1, Lines 156--202; Figure 1 and Figure 2), this causes coarse and noisy step-level signals: redundant or erroneous steps in successful trajectories may be rewarded, while useful steps in failed trajectories may be penalized. Figure 1 further shows that about 22.0\% of the steps in failed trajectories actually contribute to task progress, while about 65.3\% of the steps in successful trajectories do not meaningfully advance the task.
> > >
> > > **GraphGPO departs from trajectory-level attribution** and instead uses the unified state-transition graph to capture the global relationship (under the uniform-cost case, is reflected by the minimum-step distance to the goal) between states and the task goal. This provides a global progress signal shared across trajectories, allowing the agent to distinguish whether a step moves the state closer to or farther from success, rather than inheriting credit only from the final outcome of its own trajectory. This is precisely why the resulting step-level signal can differ substantially from trajectory-level attribution and provide more faithful supervision. We illustrate this difference in detail in Figure 2 and Figure 3.
> > >
> > > Moreover, in our further rebuttal to Reviewer 8xSD, we provide a visualization tool (at the anonymous link: https://anonymous.4open.science/r/Rebuttal_ICML_22669/Task_2276f211/Failed_Trajectory_aaa65fbe_GRPO.pdf) that directly compares how GRPO, GiGPO, and GraphGPO assign credit on ALFWorld. We believe these visualization results sufficiently justify why the cost-based signal has a substantial impact on credit assignment.
> > > ***
> > >
> > > A3: We use the form in Eq. (4) because it follows the **standard discounted-reward formulation in reinforcement learning**, where future outcomes are weighted according to their distance from the goal (Sutton \& Barto, Reinforcement Learning: An Introduction; Schulman, Proximal Policy Optimization Algorithms). This form has been widely adopted and validated in RL. In our setting, it provides a simple and monotonic way to transform graph distance into step-level reward: states closer to the goal receive larger reward, while states farther away receive smaller reward. Compared with a hard threshold or uniform reward assignment, the exponential form provides smoother and more informative supervision for credit assignment.
> > >
> > > For $\omega$, we have already provided additional discussion in Appendix D.1 (Lines 955--960). Intuitively, different choices of $\omega$ do not change the overall ranking, but they do affect the relative spacing of the resulting rewards, and thus the distribution of the normalized advantages. For example, if $d(s) \in$ \{ 1,2,3 \}, then with $\omega$=0.9, the rewards are \{0.9,0.81,0.729\}, and the corresponding normalized advantages are approximately \{1.25,-0.04,-1.20\}; with $\omega$=0.2, the rewards are \{0.2,0.04,0.008\}, and the corresponding normalized advantages are approximately \{1.40,-0.51,-0.89\}. Thus, although the ranking remains the same, the normalized advantage distribution is different, which can influence optimization dynamics in practice. Although the experimental results show that GraphGPO consistently outperforms the baselines across a range of $\omega$ values, we recommend $\omega=0.2$ in practice due to its stable performance.

---

### Official Review · Reviewer_LFci · 2026-03-13

**Soundness:** 3
**Presentation:** 3
**Significance:** 3
**Originality:** 3
**Overall Recommendation:** 4
**Confidence:** 3

**Summary:**

Existing reinforcement learning methods for LLM-based agents often suffer from "trajectory-level attribution," where every step in a successful mission is rewarded and every step in a failed one is penalized, regardless of each step's actual contribution. To address this, the authors propose GraphGPO, a novel method that aggregates multiple trial trajectories into a unified state-transition graph. By using the global structural information of this graph, the model can calculate the "distance" from any given state to the final goal and assign credit to individual steps based on whether they actually bring the agent closer to success. Experiments on benchmarks like ALFWorld, WebShop and Sokoban show that GraphGPO significantly outperforms existing methods like GRPO, achieving higher success rates with negligible computational overhead—only about 0.04% of total training time. Overall, this work offers a more precise and efficient way for AI agents to learn from both their successes and their failures.

**Compliance With Llm Reviewing Policy:**

Affirmed.

**Key Questions For Authors:**

1.How does GraphGPO handle "soft" state matching? If the environment observations are slightly noisy or stochastic, what mechanism ensures that the state-transition graph remains connected rather than forming isolated nodes?
2.Could the authors clarify if there are specific failure cases where the graph-based distance d(s) provides a misleading advantage signal, such as in "trap" states that are close to the goal in terms of steps but are actually irreversible dead ends?

**Limitations:**

Yes

**Strengths And Weaknesses:**

Strengths:
1. The paper provides a compelling data-driven motivation by quantifying the "attribution misalignment" problem. Demonstrating that 22.0% of steps in failed trajectories are actually progressive (and 65.3% in successful ones are non-progressive) provides a solid empirical foundation for the necessity of step-level credit assignment.
2. The introduction of a "critic-free" graph-based advantage estimation is a significant contribution. By leveraging the state-transition graph to derive the distance-to-goal, the framework bypasses the common instability and high resource consumption issues associated with training separate critic models in LLM-based RL.
3. The method is evaluated across diverse environments, including text-based tasks (ALFWorld, WebShop) and a VLM-based game (Sokoban), proving its generalizability. The performance gains are substantial. For instance, on the deterministic Sokoban task using Qwen2.5-VL-3B, GraphGPO achieves an 86.98% success rate, massively outperforming GRPO (67.1%) and the recent step-level method GiGPO (76.92%).

Weaknesses:
1. The core mechanism of graph aggregation relies on the exact matching of states (e.g., s_1^1=s_2^1). In complex agentic tasks with high-dimensional or long-text observations, minor variations in formatting or environment response could lead to a fragmented graph. The paper lacks a detailed sensitivity analysis on how state representation (e.g., raw text vs. embeddings) impacts the graph’s connectivity and the resulting credit assignment.
2. While GraphGPO excels in tasks with clearly defined "success nodes" (like Sokoban), its reliance on the distance-to-goal metric d(s) limits its broader applicability. It remains unclear how this framework would adapt to open-ended or qualitative tasks where a terminal "goal state" is not easily representable as a graph node.
3. Although the authors mention that graph computation is efficient, maintaining a unified state-transition graph across thousands of trajectories in environments with vast state spaces may pose memory and search-time challenges. A more rigorous discussion on the computational complexity and memory overhead as the number of training steps scales would be beneficial.
Minor issues:
P4, Section4.3, Line217: “…it indicates that the state is empirically unable to complet the task…” — Correction: It should be “complete the task”.

---

> ### Author Rebuttal · Authors · 2026-03-31
>
> Thank you for your expertise and attention to detail that has been instrumental in improving the quality of my work. Our point-by-point responses to the your mentioned questions and weaknesses are provided as follows.
>
> **Q1: (Weaknesses and Questions) The core mechanism of graph aggregation relies on the exact matching ...... graph’s connectivity and the resulting credit assignment.**
>
> A1: We agree that in high-dimensional or long-text observations, minor variations in formatting or environment responses can indeed lead to a fragmented graph. Considering a extreme case where no states can be matched, our method relies only on the trajectory-level advantage and degenerates to GRPO. Therefore, GRPO can be viewed as a lower bound of our method in the extreme case.
>
> To further examine the applicability of our method in such settings, we constructed a more challenging version of WebShop by inserting random advertisements generated by Claude. These injected advertisements act as random noise, making exact raw-text matching much more difficult and resulting in a highly sparse graph. To address this issue, we additionally used embedding-based state matching: two states are treated as identical when their embedding similarity exceeds 95\%. The experimental results are shown below.
> |Noise setting |  Raw-text matching | Embedding-based matching |
> |-------------|-----------------:|-----------------------:|
> |(q=1, p=0.3)|77.85 $\pm$3.15|**78.26$\pm$2.58**|
> |(q=1, p=0.6)|75.27$\pm$2.15|**77.42$\pm$2.38**|
> |(q=2, p=0.3)|76.85 $\pm$3.60|**78.22 $\pm$1.30**|
> |(q=2, p=0.6)|74.22$\pm$2.60|**78.26 $\pm$2.58**|
>
> Where $p$ denotes the probability of inserting advertisements into a webpage, and $q$ denotes the number of inserted advertisements. With embedding-based matching, GraphGPO remains highly robust even under severe graph sparsity and substantial observation noise.
>
> ---
>
> **Q2: (Weaknesses) While GraphGPO excels in tasks with clearly defined "success nodes" ......  where a terminal "goal state" is not easily representable as a graph node.**
>
> A2: Our work primarily focuses on RL with verifiable rewards (RLVR), which is currently a widely studied and important setting. For more open-ended tasks, group-based reinforcement learning is still at an early stage of exploration. A common practice in such settings is to use an LLM-as-a-Judge framework to evaluate final outputs and assign rewards. Since our method is mainly designed to address fine-grained credit assignment, rather than depending on a specific reward form, it can be naturally integrated with the LLM-as-a-Judge framework for open-ended tasks. Therefore, $d(s)$ does not fundamentally limit the applicability of our method.
>
> ---
>
> **Q3: (Weaknesses) Although the authors mention that graph computation is efficient ...... computational complexity and memory overhead as the number of training steps scales would be beneficial.**
>
> A3: We have discussed the memory and time overhead of our method in the **Computational Overhead** section (Line 432). For memory, the graph construction introduces essentially no additional overhead, since the environment observations used to build the graph are already stored as part of the rollout data for training. For time, Figure 5 shows that the graph-related computation accounts for only 0.04\% of each training iteration, which is almost negligible.
>
> It is also worth noting that, in the early stage of RL, the policy typically exhibits higher diversity and explores more states. As training progresses, the policy gradually concentrates on better trajectories and a more restricted state space. Therefore, rather than increasing with training steps, the computational overhead of graph-related operations tends to decrease over time.
>
> ---
>
> **Q4: (Questions) How does GraphGPO handle "soft" state matching? ...... state-transition graph remains connected rather than forming isolated nodes?**
>
> A4: In noisy, stochastic, or otherwise sparse environments, GraphGPO can incorporate embedding-based state matching to improve robustness, rather than relying solely on exact matching. **We provided additional experimental evidence for this setting in our response A1 to Q1.**
>
> ---
>
> **Q5: (Question) Could the authors clarify if there are specific failure cases ...... that are close to the goal in terms of steps but are actually irreversible dead ends?**
>
> A5: We admit that such cases can indeed arise. For example, in ALFWorld, invalid actions may produce a generic fallback observation. If this observation is mistakenly regarded in the graph as a state on the path to the goal, it may induce a misleading advantage signal and encourage the policy to revisit such dead-end-like states.
>
> This issue is mainly caused by incomplete environment information and can be alleviated by richer state tracking. In addition, the trajectory-level advantage provides a complementary signal from the overall trajectory outcome, which helps the policy avoid getting trapped in such states.

---

> > ### Author Rebuttal · Reviewer_LFci · 2026-04-04
> >
> > Thanks for the rebuttal. The response clarified several of my questions and was helpful to read. My score remains unchanged.

---

> > > ### Author Response · Authors · 2026-04-08
> > >
> > > We sincerely thank the reviewer for the time and effort devoted to our work. We are grateful that our response helped clarify several of the reviewer’s questions, and we truly appreciate the reviewer’s continued positive evaluation of our work. Should these clarifications further improve the reviewer’s overall assessment of our contribution, we would be sincerely grateful for any additional consideration.

---

### Decision · Program_Chairs · 2026-04-30

**Decision:**

Accept (regular)

**Comment:**

This work introduces Graph-based Group Policy Optimization (GraphGPO), a framework that assigns step-level learning signals by aggregating rollout trajectories into a unified state-transition graph and estimating the distance from each state to the task goal.

The Strengths of this work are listed as follows.
1. Step-level credit assignment in RL for LLM agents is an important challenge.
2. The researcher applies graph-based advantage estimation into LLM agent training.
3. Good performance, theoretical proofs, well written, and easy to follow.

The major weakness of this work are:
1. Exact state matching may cause graph fragmentation in noisy environments.
2. Its adaptability to open-ended tasks without clear goals is questionable.
3. The confusing presentation of the cost function raises fairness concerns.

The strengths out-weight the weakness, please take into account the reviews when preparing revisions.